# Tandem dehydrogenation-olefination-decarboxylation of cycloalkyl carboxylic acids via multifold C−H activation

Tanay Pal[1], Premananda Ghosh[1,2], Minhajul Islam[1,2], Srimanta Guin[1], Suman Maji[1], Suparna Dutta [1], Jayabrata Das [1], Haibo Ge [3] ✉ & Debabrata Maiti [1,2] ✉

Dehydrogenation chemistry has long been established as a fundamental aspect of organic synthesis, commonly encountered in carbonyl compounds. Transition metal catalysis revolutionized it, with strategies like transfer-dehydrogenation, single electron transfer and C−H activation. These approaches, extended to multiple dehydrogenations, can lead to aromatization. Dehydrogenative transformations of aliphatic carboxylic acids pose challenges, yet engineered ligands and metal catalysis can initiate dehydrogenation via C−H activation, though outcomes vary based on substrate structures. Herein, we have developed a catalytic system enabling cyclohexane carboxylic acids to undergo multifold C−H activation to furnish olefinated arenes, bypassing lactone formation. This showcases unique reactivity in aliphatic carboxylic acids, involving tandem dehydrogenation-olefination-decarboxylation-aromatization sequences, validated by control experiments and key intermediate isolation. For cyclopentane carboxylic acids, reluctant to aromatization, the catalytic system facilitates controlled dehydrogenation, providing difunctionalized cyclopentenes through tandem dehydrogenation-olefination-decarboxylation-allylic acyloxylation sequences. This transformation expands carboxylic acids into diverse molecular entities with wide applications, underscoring its importance.

The unsaturation in an organic molecule creates multifaceted opportunities to diversify it in a desirable manner, which makes the genre of dehydrogenation/unsaturation chemistry an important aspect in organic synthesis. The traditional pathway to generate the unsaturation in carbonyl compounds involved the enolate or enamine chemistry (Fig. 1a)[1–11]. Later, the classical enolate chemistry was superseded by transition metal catalysis. In this regard, the C−H activation had been a prevailing pathway to initiate the dehydrogenation either through transfer dehydrogenation process[12–14] or β-hydride elimination route[15–25]. There are also precedences of radical involvement in dehydrogenation processes promoted by the merger of metal catalysis with photocatalysis or electrocatalysis[26–32]. While ketones are favorable

choices as substrates for dehydrogenation, the aliphatic carboxylic acids in its native form offer more challenges for generating unsaturation in a molecule. The aliphatic carboxylic acid in its masked enediolates form are amenable to be oxidized to α,β-unsaturated acid by Pd(II) complexes[33]. Also Pd(II) in conjunction with pyridine-pyridone-based ligand enables divergent dehydrogenation of carboxylic acids driven by the β-methylene C−H activation. The ligand control helps in inhibiting the over-reaction of the dehydrogenated product by subsequent C−H activation[34].

The above discussed protocols are relevant for the generation of single unsaturation, however, there are provisions for generating multiple unsaturation in a molecule through subsequent

[1]Department of Chemistry, Indian Institute of Technology Bombay, Mumbai, India. [2]IITB-Monash Research Academy, Indian Institute of Technology Bombay, Mumbai, India. [3]Department of Chemistry and Biochemistry, Texas Tech University, Lubbock, USA. ✉e-mail: haibo.ge@ttu.edu; dmaiti@iitb.ac.in

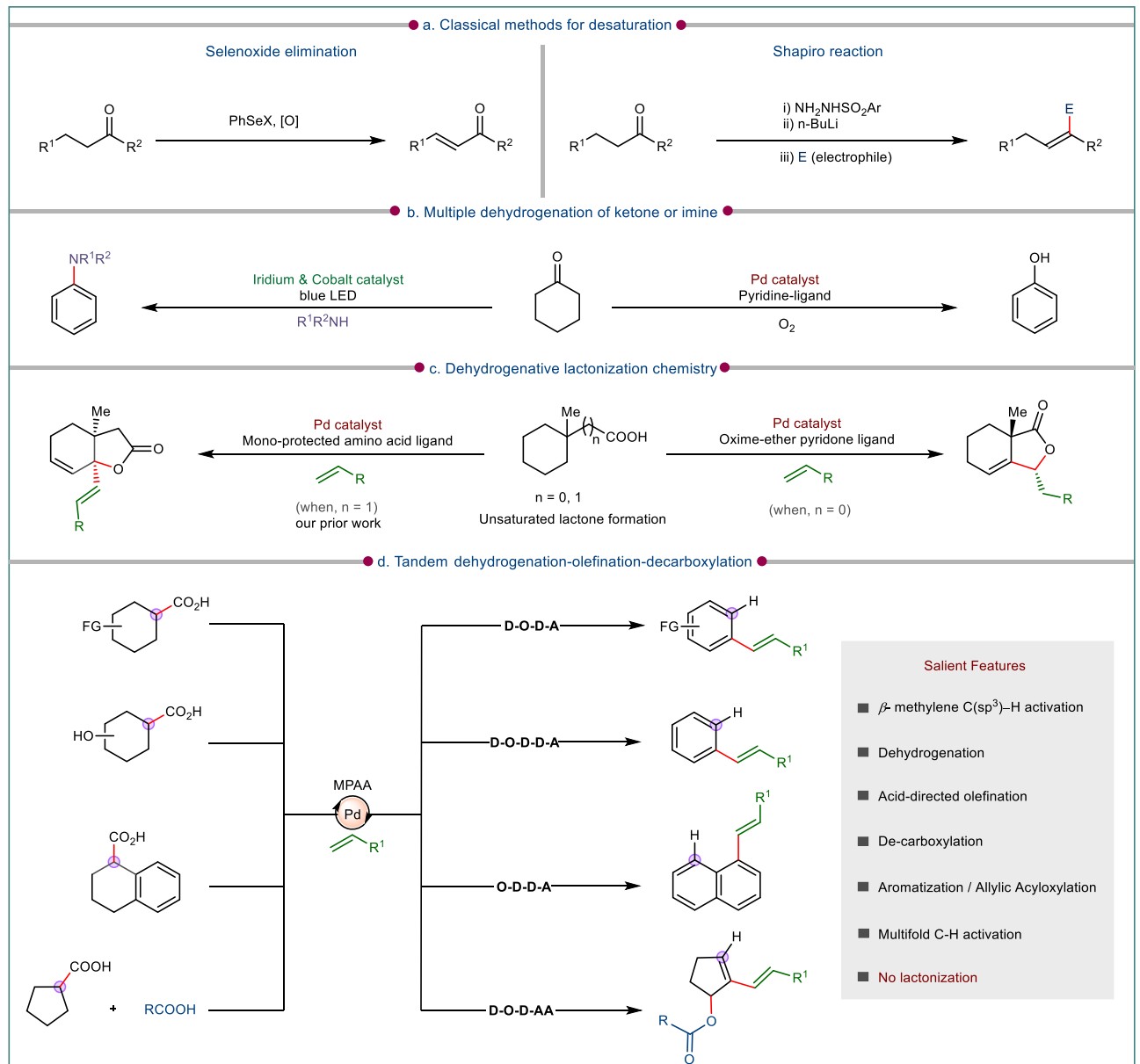

**Fig. 1 | Evolution and advancements in dehydrogenation strategies of organic molecules. a** Traditional approaches to achieve unsaturation. **b** Strategies towards metal catalyzed dehydrogenation in cyclohexanones leading to aromatization. **c** Trends of reactivity in homologous cyclic acids. **d** This work: One pot dehydrogenation and de-carboxylative functionalization via multifold C−H activation

(MPAA: mono-protected amino acid, D-O-D-A: dehydrogenation-olefination-decarboxylation-aromatization, D-O-D-D-A: dehydrogenation-olefination-decarboxylation-dehydroxylation-aromatization, O-D-D-A: olefination-dehydrogenation-decarboxylation-aromatization, D-O-D-AA: dehydrogenation-olefination-decarboxylation-allylic acyloxylation).

dehydrogenation. This phenomenon is more common with the cyclic ketones as precursors in which a relayed dehydrogenation initiated by C−H activation or through single electron transfer process leads to an aromatic molecule in the form of phenol, anilines or ethers (Fig. 1b)[35–45]. Such dehydrogenative aromatization processes are beneficial in providing certain substituted arenes which are otherwise difficult to access via traditional routes. In a very recent report, it has been observed that Pd/ligand combination enabled the conjugated diene formation in linear aliphatic carboxylic acids and cyclohexyl carboxylic acid provided benzoic acid derivatives[46]. While these methodology developments entail about the creation of double bonds only, a tandem dehydrogenation-functionalization or vice-versa is another facet of C−H activation reactions that are elusive in nature but affords products of high value. This has been witnessed with aliphatic carboxylic acids which upon reaction with alkynyl bromides gives γ-

alkylidene butenolides[34]. Further, with α-substituted carboxylic acids, a tandem dehydrogenation−olefination−lactonization affords β-alkylidene-γ-lactones (Fig. 1c)[47]. In a recently developed protocol from our group, we have observed a very distinct reactivity of cycloalkyl acetic acid having a β-substituent. This substrate class in presence of Pd/ligand combination presented a unique reactivity of overriding the usual reactivity mode to form bicyclic lactones bearing an allylic double bond. The same substrate in presence of olefin and allyl alcohols led to an all-carbon quaternary center at the ring junction (Fig. 1c)[48]. Hence a slight change in the substrate class led to different classes of products.

In this work, we showcase the reactivity of cycloalkyl carboxylic acids with olefin under a very similar condition to that applied for cycloalkyl acetic acid[48]. Intriguingly, the cyclohexyl carboxylic acids exhibit a unique reactivity to be transformed to olefinated arenes via

multiple dehydrogenation (Fig. 1d). It may be referred here that with cyclohexyl carboxylic acid bearing an α-substituent, a lactone product was observed along with dehydrogenation at the β,γ position[47]. Removing the α-substituent confers a distinct identity to cyclohexyl carboxylic acid by a tandem dehydrogenation-olefination-decarboxylation-aromatization (D-O-D-A) sequence. In cases of the smaller cycloalkyl systems, where aromatization is not possible, the present transformation gives difunctionalized cycloalkenes (Fig. 1d). The reaction sequence for such substrates follow dehydrogenation-olefination-decarboxylation-allylic acyloxylation (D-O-D-AA). While the early stages of C−H activation focused on single point functionalization, with the inception of multiple C−H activation in a single molecule and unlocking differential reactivity modes of a substrate with the assistance of ligands, this domain is expected to usher in the next phase of organic synthesis and contribute to further advancement in creating valued products.

## Results

### Optimization of the reaction conditions

The endeavor towards investigating the reactivity of cycloalkyl carboxylic acid commenced with the reaction between 4-$^t$Bu-cyclohexyl carboxylic acid and ethyl acrylate. In all our prior Pd-catalyzed C−H functionalizations of aliphatic carboxylic acids, we have observed that mono-protected amino acid (MPAA) is indispensable to enable the functionalization[48,49]. A base usually in the form of Na salt is a pivotal requirement to bind the acid functionality in $\kappa_1$ mode that allowed facile C−H activation. Accordingly, the reaction condition comprising of 10 mol% Pd(OAc)$_2$, 20 mol% N-Ac-Leu-OH, 2 equiv. Ag$_2$CO$_3$, 2 equiv. Na$_2$HPO$_4$ in HFIP solvent under air at 110 °C was chosen for the present transformation; similar to our recent report on the bicyclic lactone formation from cycloalkyl acetic acid[48]. Surprisingly, the initial reaction provided an olefinated arene **1** as the major product via multifold dehydrogenation-decarboxylation path, while a bis-dehydrogenated lactone product was obtained as the minor product (see Supplementary Information, Supplementary Table 1).

Under the same conditions, various Pd salts afforded different ratios of these products but with π-allyl and π-cinnamyl Pd(II) complexes, the minor lactone product could be completely eliminated to provide the olefinated arene as the only product. Considering yields, the π-cinnamyl Pd(II) complex was superior over the π-allyl complex. Although [Pd$_2$(dba)$_3$] provided higher yield (58%) than [Pd(π-cinnamyl)Cl]$_2$ (53%), we continued with [Pd(π-cinnamyl)Cl]$_2$ as it did not give any side product, unlike the case of [Pd$_2$(dba)$_3$] (see Supplementary Information, Supplementary Table 1). Further, a series of N-protected α-amino acids were studied and it was observed that the inclusion of N-Ac-Val-OH as a ligand improved the yield of **1**. The pyridone and pyridine-based ligands which are widely used for C(sp³)−H functionalizations were not much effective in the present transformation. The optimization of other parameters such as oxidant, base, solvent, and temperature revealed that the ones used for the preliminary reaction are ideal. Deviation from those parameters hampered the product yield. Only prolonging the reaction time from 24 to 36 h was beneficial in providing the highest achievable yield of **1**. The final reaction conditions constitute of [Pd(π-cinnamyl)Cl]$_2$ (10 mol%), N-Ac-Val-OH (20 mol%), Ag$_2$CO$_3$ (2 equiv.), Na$_2$HPO$_4$ (2 equiv.), HFIP (hexafluoroisopropanol) as solvent at 110 °C for 36 h (under air), which were found suitable to execute the tandem D-O-D-A of cyclohexyl carboxylic acid and afford the olefinated arene **1**. It is noteworthy to mention that, the C−H olefination of aliphatic carboxylic acid is always associated by lactone formation[47–56]. However, this is an unexplored realm where the lactone formation could be bypassed to an alternative decarboxylation pathway by selective catalyst screening. The details of the optimization studies are provided in the supplementary information (for detailed optimization studies, see Supplementary Information, Section 2.1. Optimization Details).

## Substrate scope

We next evaluated the D-O-D-A reactivity for a series of cyclohexyl carboxylic acids under the optimized conditions (Fig. 2). With the unsubstituted cyclohexyl carboxylic acid, the transformation provided decent yield of the respective olefinated benzene **2**. Next, the protocol was applied to alkyl substituted cyclohexyl carboxylic acids. The 2-Me cyclohexyl carboxylic acid afforded low yield of the 3-Me substituted olefinated benzene **3**, owing to the steric hindrance that affects the C−H activation. For the 3-Me substituted cyclohexanoic acid having an opportunity of C−H activation at either β and β′ position of the carboxylic acid functionality is expected to lead a pair of regioisomeric olefinated products. However, the activation occurred preferentially only at the C−H position distal to the Me substituent to afford solely **4**. The usual D-O-D-A reactivity was observed with the 4-Me substituted cyclohexanoic acid to give the olefinated arene **3**. Hence, **3** can be obtained from both 2- and 4-Me substituted cyclohexanoic acid. However, with 4-Me substrate the yield was better. The protocol thus allows for a choice of the proper substrate to avail a better yield of a particular regioisomer. A range of other 4-alkyl substituted cyclohexyl carboxylic acid underwent smooth transformations affording the respective meta-olefinated arenes (**5–9**) in synthetically useful yields. Thus, this protocol offers an alternate route to form *meta* or *para*-olefinated arenes where commercially available 4- and 3-substituted cyclohexane carboxylic acids could be utilized as starting materials. The reactions of 4-cycloalkyl substituted cyclohexanoic acid were also amenable to the D-O-D-A sequence, giving moderate to good yields of their respective products (**10–11**). In all the alkyl substituted cyclohexanoic acid it was observed that the dehydrogenation remained confined to the cyclohexyl moiety possessing the carboxylic functionality and not relayed to the alkyl substituents. Other functionalities at the 4-position of cyclohexyl carboxylic acids such as ether (**12**), ester (**13**), trifluoromethyl (**14**) and phenyl (**15**) were all tolerated under the present reaction conditions. The tolerance of ester group in **13** can serve as the point of utilization for subsequent derivatization of the arene. The transformation executed on the phenol tethered substrate exhibited the same reactivity to give olefinated arene (**16**). Also, 4-ethylphenol, a botanical agrochemical[57], tethered with cyclohexane carboxylic acid reacted efficiently to provide corresponding olefinated product (**17**). Further, the robustness of the protocol was demonstrated by generating structurally diverse molecules from the cyclohexyl substrates with phenolic natural product pendants viz. thymol (**18**), carvacrol (**19**), and propofol (**20**) which provided the respective olefinated products in acceptable yields without any compromise in reactivities (Fig. 2).

The reactivity of 1,2,3,4-tetrahydro-1-naphthoic acid with ethyl acrylate was ventured next (Fig. 2). 1,2,3,4-Tetrahydro-1-naphthoic acid is a fused bicyclic substrate which has the provision for C−H activation at both C2- and C8-positions. Hence the olefin counterpart can be coupled either at the C2-position (which is the usual reactivity observed so far with the cyclohexyl carboxylic acid) or at the C8-position. In this case, olefination took place at the C8-position (**21**) suggesting that acid directed olefination occurred initially, followed by decarboxylation and aromatization. However, if the C8-position is blocked by another group, then the olefin was inserted at the C2-position (**22**) by usual D-O-D-A sequence; albeit other regio-isomeric olefinated products formation were observed. Further, this substrate underwent a dehydrogenation-decarboxylation-aromatization (D-D-A) sequence to form naphthalene derivative (**22′**).

Next, we sought to examine 1-substituted cyclohexane carboxylic acids. Despite having an α-quaternary center, the decarboxylation facilitated the aromatization process in such acid substrate classes. It had been observed that 1-methyl and 1-propyl substituted cyclohexane carboxylic acids underwent to D-O-D-A sequence yielding 2-alkyl substituted olefinated benzene (**23–24**). However, minor amount of gamma-lactones were also formed as side

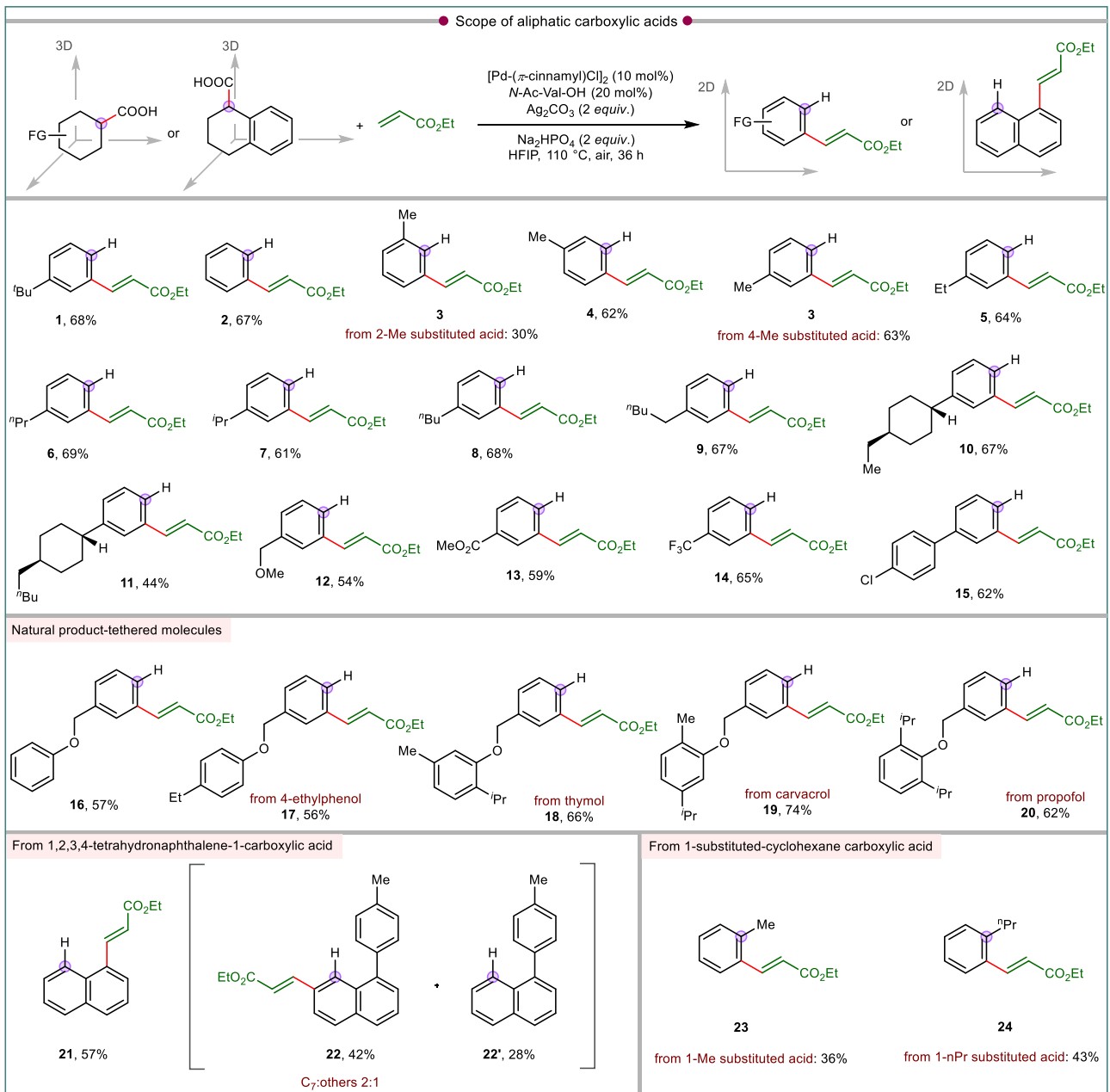

**Fig. 2 | Unique reactivity modes of cyclohexane carboxylic acids: scope of aliphatic carboxylic acids.** All reactions were conducted at 0.1 mmol scale and isolated yields are reported. Standard condition: acid (0.1 mmol), olefin (2 *equiv.*), [Pd($\pi$-cinnamyl)Cl]$_2$ (10 mol%), N-Ac-Val-OH (20 mol%), Ag$_2$CO$_3$ (2 *equiv.*), Na$_2$HPO$_4$ (2 *equiv.*), HFIP (1 mL), 110 °C, air, 36 h.

products in these cases (see, Supplementary Information, Entries **23'** and **24'**).

Subsequently, the scope of activated olefins as coupling partners was explored employing the optimal conditions (Fig. 3a). An array of diversely substituted acrylates including alkyl (**25**–**26**), cyclohexyl (**27**), benzyl (**28**–**30**), perfluoroalkyl (**31**) were effective coupling partners in this transformation to give the desired olefinated products in moderate to good yields, irrespective of the nature of carboxylic acids. The cyclic $\alpha,\beta$-unsaturated esters led to the formation of allylated arene derivatives (**32**–**33**) instead of the vinylated arene products. The occurrence of such allylation can be explained by the fact that, in the intermediate containing cyclic tri-substituted olefins, syn-$\beta$-hydride elimination is only facile from a site favoring the production of allylated products[58,59]. Apart from the acrylate derivatives, the present transformation was equally efficient with activated olefins possessing

ketone (**34**), nitrile (**35**–**36**), amide (**37**–**41**), phosphonate (**42**–**44**) and sulfone (**45**). These activated olefins afforded corresponding olefinated arenes via tandem D-O-D-A sequence in good yields. Dimethyl fumarate, an internal olefin is deemed as a challenging substrate in olefination. However, the compatibility of this internal olefin under the present catalytic system was showcased by the formation of the corresponding olefinated product (**46**) in 40% yield with a *Z:E* ratio of 10:1. While activated olefins are preferred choices as coupling partners in C(sp$^3$)–H activation reactions, they often fail to accommodate semi-activated styrene substrates. In fact, there are no reports known for employing styrenes as olefin partner in carboxylate mediated C(sp$^3$)–H activation. The current protocol could be extended to a series of styrene substrates, the reactivity is governed by the kind of substitution present in the phenyl ring (Fig. 3b). Pentafluoro styrene with 4-$^t$Bu cyclohexane carboxylic acid provided excellent yield of 83% for the

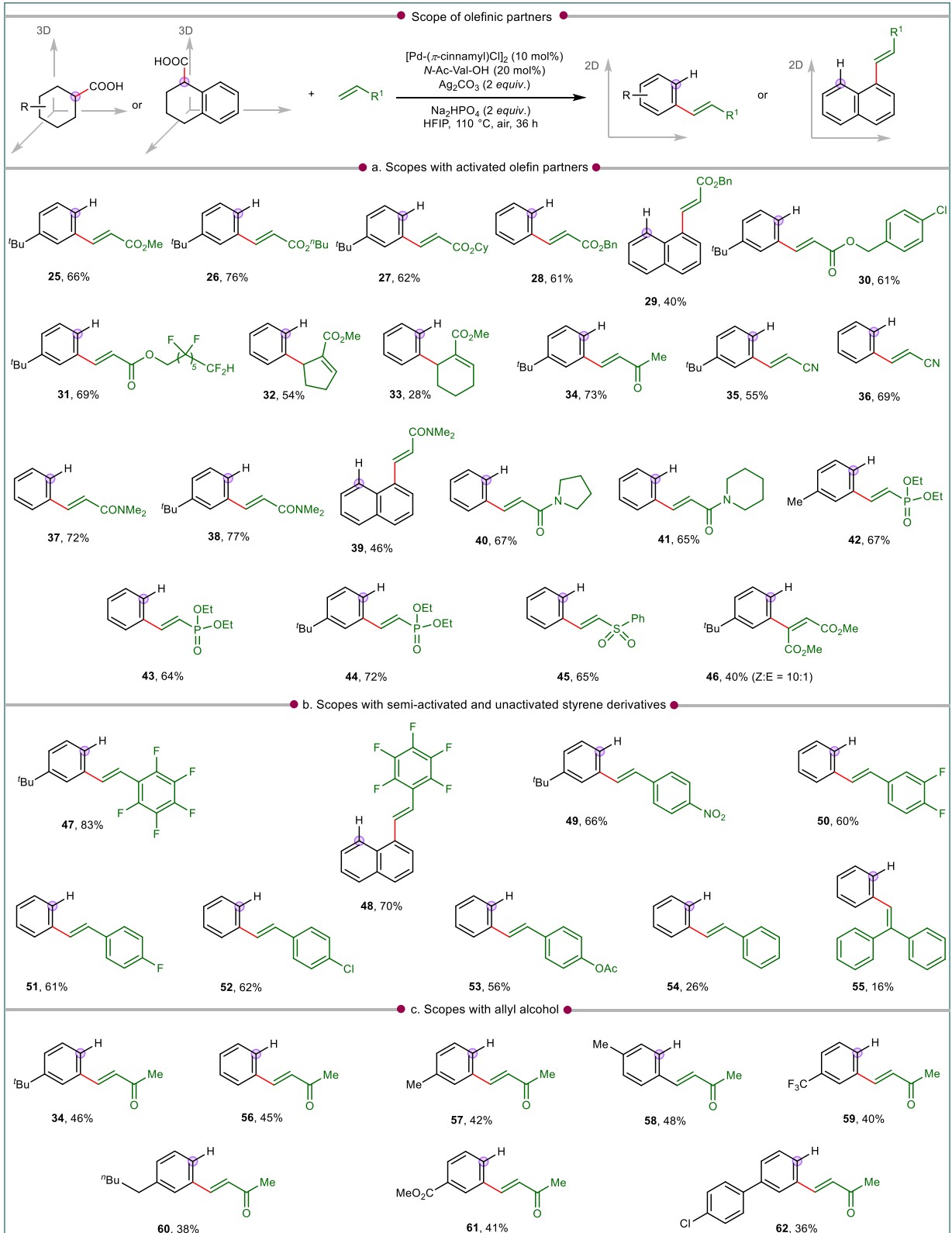

**Fig. 3 | Diversification of activated and semi-activated olefinic partners.** Scopes of the current de-carboxylative aromatization protocol with **a** activated olefins, **b** semi-activated and unactivated olefins, **c** allyl alcohols. All reactions were conducted at 0.1 mmol scale and isolated yields are reported. Standard condition: acid (0.1 mmol), olefin (2 *equiv.*), [Pd(π-cinnamyl)Cl]₂ (10 mol%), *N*-Ac-Val-OH (20 mol%), Ag₂CO₃ (2 *equiv.*), Na₂HPO₄ (2 *equiv.*), HFIP (1 mL), 110 °C, air, 36 h.

unsymmetrically substituted stilbene derivative (**47**). High yields of the respective stilbene derivatives were obtained for the reaction with 1,2,3,4-tetrahydro-1-naphthoic acid (**48**). Other than pentafluoro styrene, 4-nitro styrene (**49**), 3,4-difluoro styrene (**50**), 4-fluorostyrene (**51**), 4-chloro styrene (**52**) and 4-acetoxy styrene (**53**) reacted efficiently with cyclohexyl carboxylic acids to form corresponding unsymmetrically substituted stilbene derivatives with a range of 56 to 66% yields. Styrene and 1,2-gem-diphenylethylene were also found to

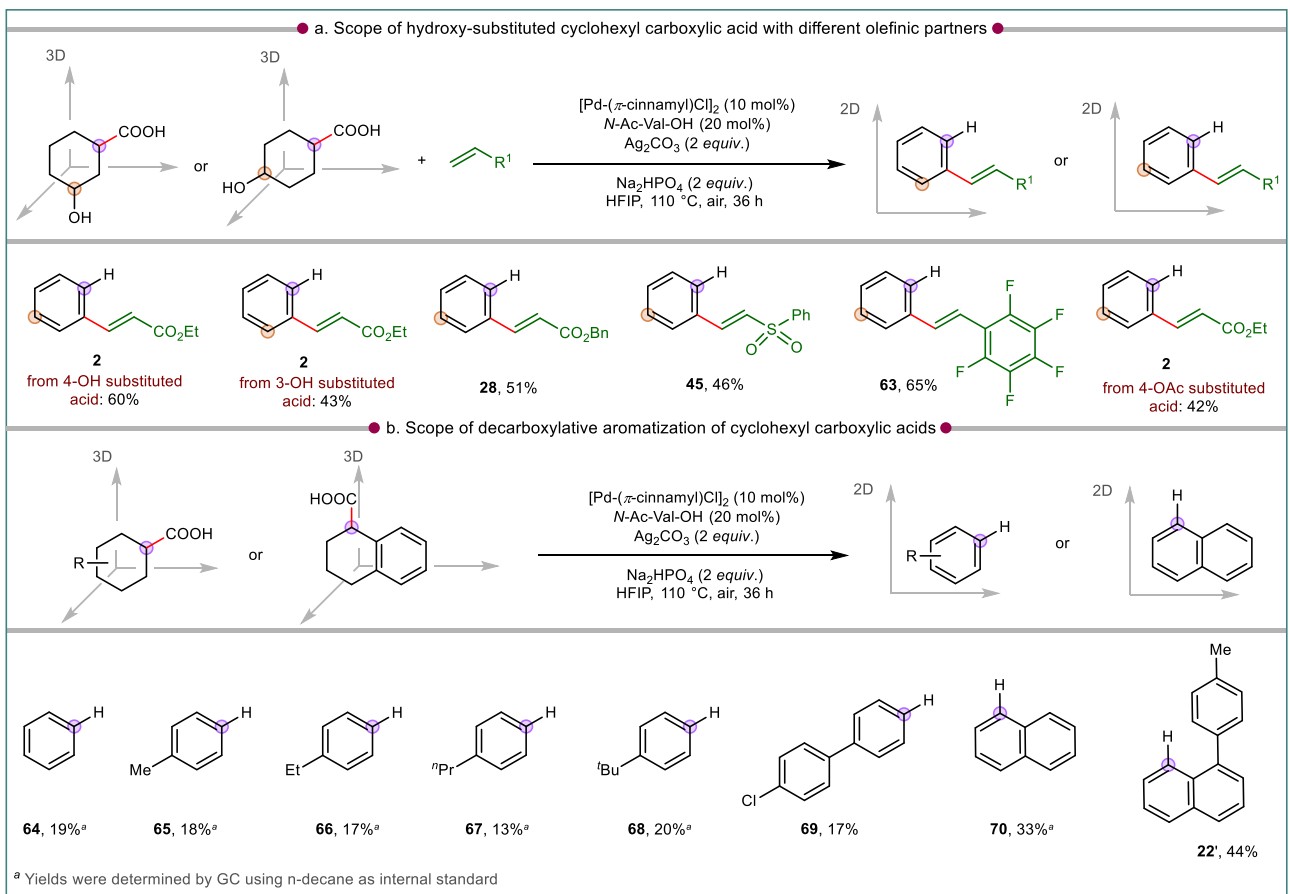

**Fig. 4 | Strategies for D-O-D-D-A and D-D-A.** All reactions were conducted at 0.1 mmol scale and isolated yields or GC yields (mentioned) are reported. Standard condition: acid (0.1 mmol), olefin (2 *equiv.* for (**a**), not being used for without coupling partner approach (**b**), [Pd(π-cinnamyl)Cl]₂ (10 mol%), *N*-Ac-Val-OH (20 mol%), Ag₂CO₃ (2 *equiv.*), Na₂HPO₄ (2 *equiv.*), HFIP (1 mL), 110 °C, air, 36 h.

be amenable under the reaction conditions to produce trans-stilbene (**54**) and triphenylethylene (**55**) albeit their reactivity dropped to give lower yields. Another class of olefins that could be encompassed under the present transformation is allyl alcohol (Fig. 3c). Though allyl alcohols are well known as coupling partner in C(sp²)−H activations, reports for its practical applications in the field of unactivated C(sp³)−H functionalization remain scarce[48,56]. However, the successful utilization of allyl alcohols as olefinic partner with cyclohexyl carboxylic acids (with or without substitution) under the optimized conditions gave access to α,β-unsaturated ketones incorporated arenes in moderate yields (**34**, **56−62**, Fig. 3c).

We were curious to inspect the reactivity of 4-hydroxy-substituted cyclohexyl carboxylic acid under the present reaction conditions. If the substrate reacts via the usual D-O-D-A pathway, then a phenol derivative was expected while the presence of free -OH group could also poison the catalyst and inhibit the reaction. Intriguingly, all the above possibilities were over-ridden and a de-hydroxylation pathway[60–62] took over and resulted in the olefinated arene **2** (Fig. 4a). Analogous observation was also made with the 3-hydroxy substituted cyclohexanoic acid which provided the same olefinated product **2**. Hence, these substrates underwent three different de-functionalizations: dehydrogenation-dehydroxylation-decarboxylation which is a rare event in C−H activation-initiated transformations (Fig. 4a). The same reactivity of hydroxy substituted cyclohexane carboxylic acid prevailed for its reaction with other activated olefins (**28**, **45**) and styrene (**63**). A comparable reactivity involving acetoxy elimination[61,62] was observed when 4-acetoxy-substituted cyclohexyl carboxylic acid underwent olefination and decarboxylative aromatization, leading to the formation of ethyl cinnamate (**2**).

In absence of olefin coupling partner, cyclohexane carboxylic acids undergo a decarboxylation-aromatization sequence to give arenes (**64−70, 22'**, Fig. 4b), albeit in low yields. Only for 1,2,3,4-tetrahydro-1-naphthoic acids, the yields for the corresponding naphthalene product (**70, 22'**) were slightly higher since the aromaticity drives the decarboxylation in this case. This observation varies from a very recent report where only dehydrogenation occurred to provide benzoic acids[46]. Hence, the reactivity depends on the catalytic system employed for a transformation.

The reactivity of cyclohexyl carboxylic acid or its analogs were studied initially under the optimized conditions; all of which followed the D-O-D-A sequence preferentially to give the olefinated arenes. However, our aim was to study the behavior of cycloalkyl acids for which aromatization is not possible. Towards this goal, cyclopentyl carboxylic acid was chosen as the substrate of interest and executed to the present transformation with ethyl acrylate. Interestingly, the reaction led to a decarboxylated difunctionalized cyclopentene derivative (**71**) with olefin being introduced at the vinylic position and the cyclopentyl carboxylic acid itself is incorporated at the allylic position (Fig. 5a). The yield was moderate considering that acid substrate itself undergoes self-coupling. The same pathway was followed upon reaction of the same cyclopentyl carboxylic acid with a set of other acrylates and methyl vinyl ketone (**72−76**, Fig. 5a). Following the fact that acid itself is a coupling partner in this reaction we wanted to introduce other acids in the reaction to see if they can effectively couple at the allylic position of cyclopentene. In presence of other cycloalkyl acids which are not prone or less susceptible to aromatization under present conditions, a tandem multicomponent reaction occurred in which the other acid partner underwent efficient cross-coupling

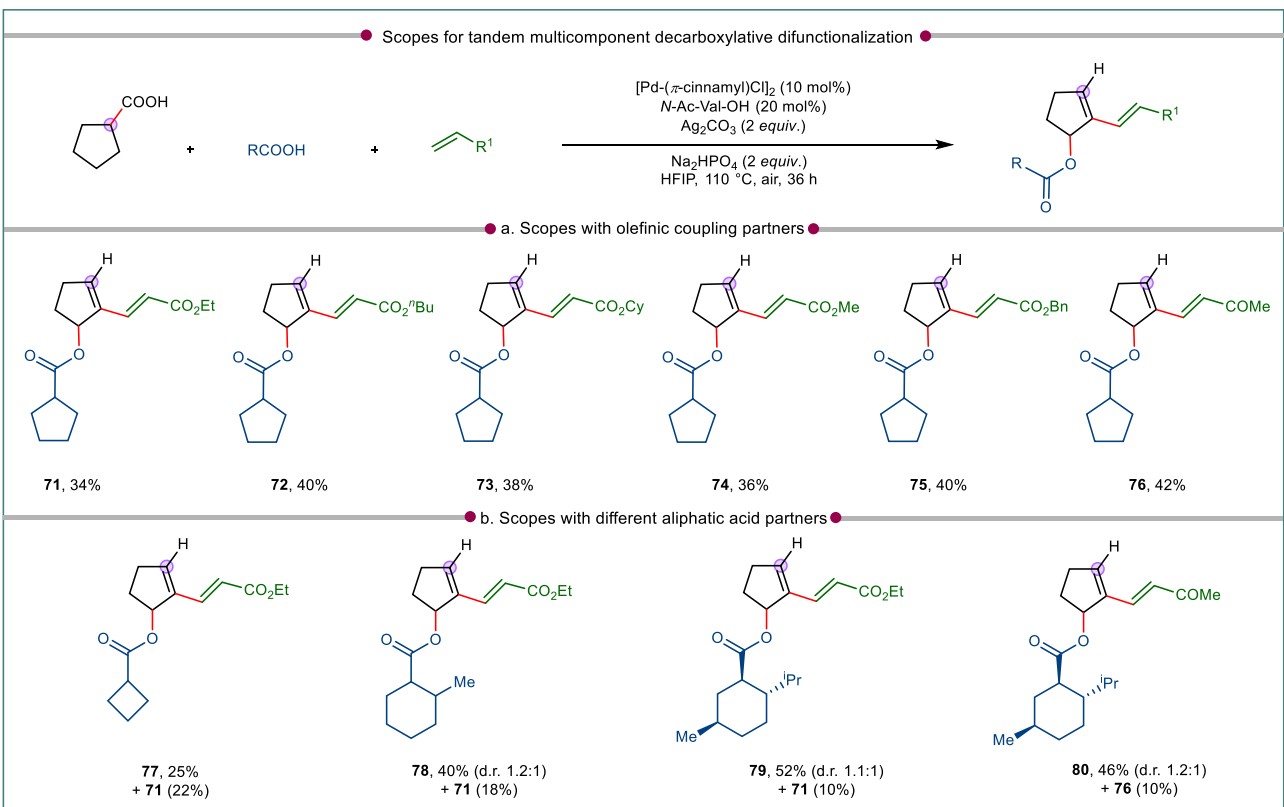

**Fig. 5 | Unified strategy for synthesis of difunctionalized cyclopentene derivatives via controlled dehydrogenation.** Scopes of the established multicomponent reaction protocol with **a** diverse olefinic coupling partners, and **b** aliphatic acid partners. All reactions were conducted at 0.1 mmol scale and isolated yields are reported. Standard condition: cyclopentane carboxylic acid (0.1 mmol), cycloalkyl carboxylic acid (4 to 6 membered cycloalkyl carboxylic acid, 1 *equiv.*), olefin (2 *equiv.*), [Pd($\pi$-cinnamyl)Cl]$_2$ (10 mol%), *N*-Ac-Val-OH (20 mol%), Ag$_2$CO$_3$ (2 *equiv.*), Na$_2$HPO$_4$ (2 *equiv.*), HFIP (1 mL), 110 °C, air, 36 h.

to give the difunctionalized cyclopentene products (**77**–**80**, Fig. 5b). However, along with the formation of hetero coupling products some minor amount of self-coupled product (e.g., **71**, **76**) also formed (Fig. 5b).

## Applications
The development of tandem D-O-D-A sequence gives ample opportunity to utilize such a protocol for direct synthesis of bio-relevant molecules of pharmaceutical interest or their precursors. The developed catalytic system expedited the synthesis of late-stage regiospecific olefinated monobenzone derivative, used for the treatment of vitiligo (**81**, Fig. 6a)[63,64]. The protocol thus allows to conduct a late-stage functionalization without actually resorting to the actual drug molecule which would otherwise have given a mixture of regioisomers upon direct olefination. The reaction of 3-methyl cyclohexyl carboxylic acid and acryl amide gave the respective olefinated arene (**82**, Fig. 6b), the reduction of which generated the precursor **83** for the synthesis of triprolidine, an antihistamine drug (Fig. 6b)[65,66]. The vinylic ester **33** on hydrolysis gave the corresponding carboxylic acid **84** which when treated with triflic acid resulted in the formation of the tetrahydrofluorenone derivative **85** (Fig. 6c). This compound could have a potential pharmaceutical significance having antifungal activity[67]. Also, such tetrahydro-fluorenone can serve as precursor for the synthesis of pharmaceutically relevant benzohopane (Fig. 6c)[68]. The vinylic acid **86** obtained by hydrolysis of vinyl ester **32** was subjected to further diversification by Rh catalyzed $\beta$-olefination of the corresponding acid functionality, thus resulting in a tri-functionalized cyclopentene derivative **87** (Fig. 6d)[69]. The olefinated arene **16** upon treatment with BBr$_3$ led to deprotection of ether and ester, to give quantitative yield of meta-olefinated benzyl bromide (**88**, Fig. 6e). It is noted that Heck

reaction of benzyl bromides is difficult to conduct, hence to avail the olefinated benzyl bromide this can be utilized as an alternative route.

## Mechanistic investigations
The structural pattern of the product formed with cyclohexyl carboxylic acid indicates that the initiation of C−H activation is followed by three fundamental events: dehydrogenation, olefination, and decarboxylation. In order to delve into the mechanistic intricacies of this transformation it is important to probe the sequence of the aforementioned events. With our prior knowledge on dehydrogenative C−H-activation in aliphatic carboxylic acid, we hypothesized multiple routes which could be feasible to enable the product formation. We conducted a series of control experiments and isolated various intermediates to validate the reaction pathway for this transformation. The occurrence of dehydrogenation by $\beta$-hydride elimination is most likely the initial step for this transformation following C−H activation. However, there are two possibilities for dehydrogenation which could either lead to an $\alpha,\beta$-unsaturation (Int-B) or the $\beta,\gamma$-unsaturation (Int-B'). Since, the $\alpha,\beta$-unsaturated carboxylic acid (Int-B) is thermodynamically more stable than its regioisomer Int-B', Int-B was considered for further investigation. The Int-B can subsequently undergo aromatization followed by olefination or vice-versa. As per the former sequence, aromatization will first lead to benzoic acid (Int-Ar) which will then undergo ortho-olefination of benzoic acid followed by decarboxylation. To check the viability of this hypothesized route, benzoic acid (Int-Ar) had been subjected to reaction with an olefinic coupling partner under the standard conditions. The reaction afforded a lactone derivative (**89**, Fig. 7a) rather than the desired olefinated arene **2**. This suggests that benzoic acid is not an intermediate and this pathway is unlikely. An alternate sequence is where the olefination precedes the

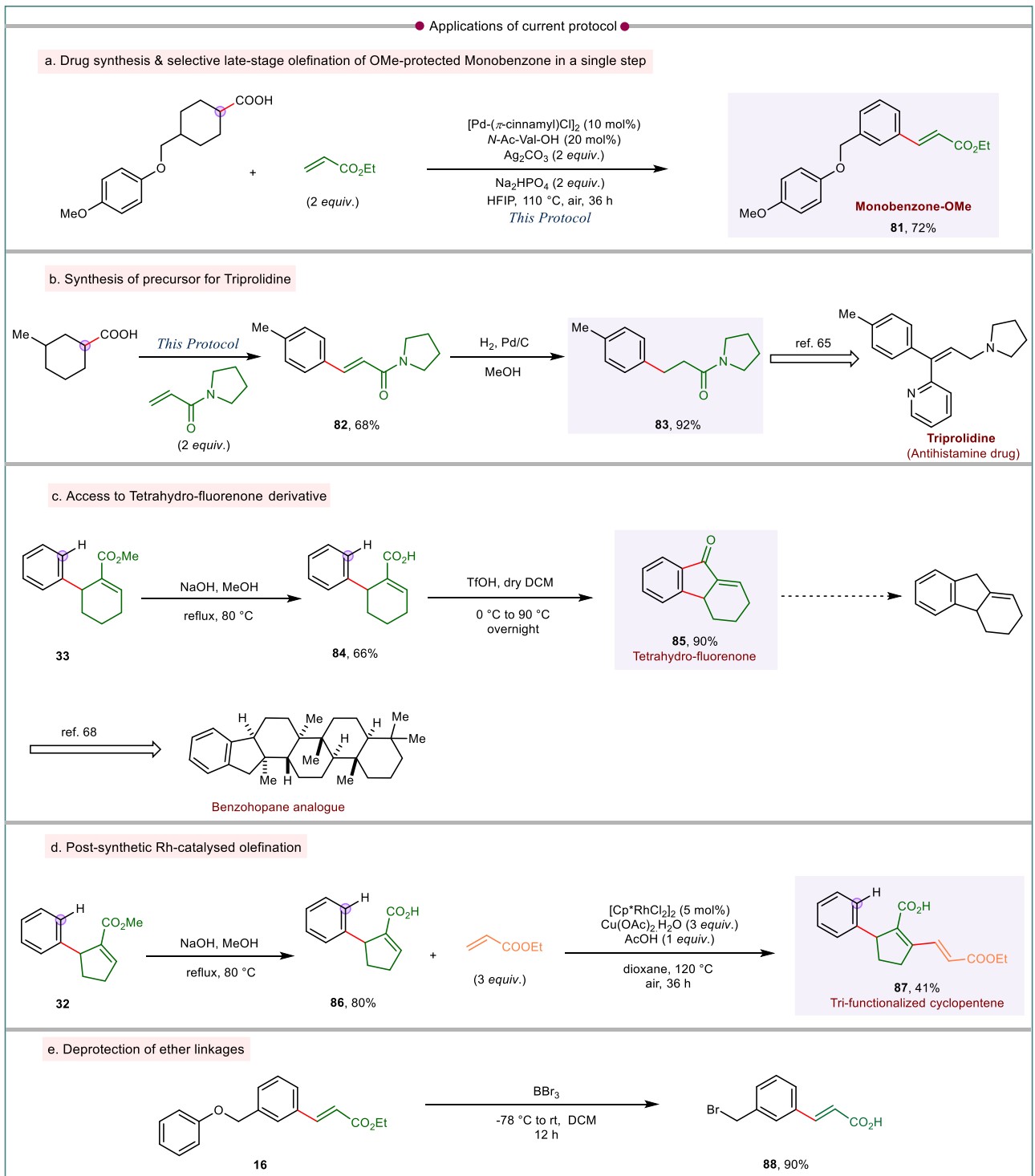

**Fig. 6 | Applicability and post-synthetic modification of current protocol.**
**a** Expedited synthesis of monobenzone derivative with exclusive regioselectivity.
**b** Synthesis of triprolidine synthon. **c** Access to tetrahydro-fluorenone, an
important building block for benzohopane analog. **d** Synthesis of trifunctionalized
cyclopentene derivative. **e** Post-synthetic deprotection strategy to synthesize meta-
olefinated benzyl bromide.

aromatization. To examine the possibility of this route, cyclohex-1-ene-
1-carboxylic acid (Int-B) was subjected to standard reaction conditions
with ethyl acrylate as coupling partner. However, the α,β-unsaturated
cyclohexene carboxylic acid did not produce our desired olefinated-
aryl product **2**, albeit furnished a lactone derivative (**90**, Fig. 7b) with
74% yield. This proves that cyclohex-1-ene-1-carboxylic acid (Int-B) is
not an intermediate involved in the mechanistic cycle and it is highly
probable that the reaction proceeds via Int-B'. Similar to Int-B, the

same reaction sequence is also possible with Int-B'. Since benzoic acid
did not-afford the olefinated arene, the previous argument that aro-
matization does not precede olefination holds good for Int-B' as well.
An alternative possibility following the formation of Int-B' may involve
early-stage olefination and subsequent aromatization in the penulti-
mate stages. To assess the feasibility of the suggested reaction
sequence, we conducted experiments with the pre-synthesized β,γ-
unsaturated cyclohexene carboxylic acid (Int-B'). To our satisfaction, it

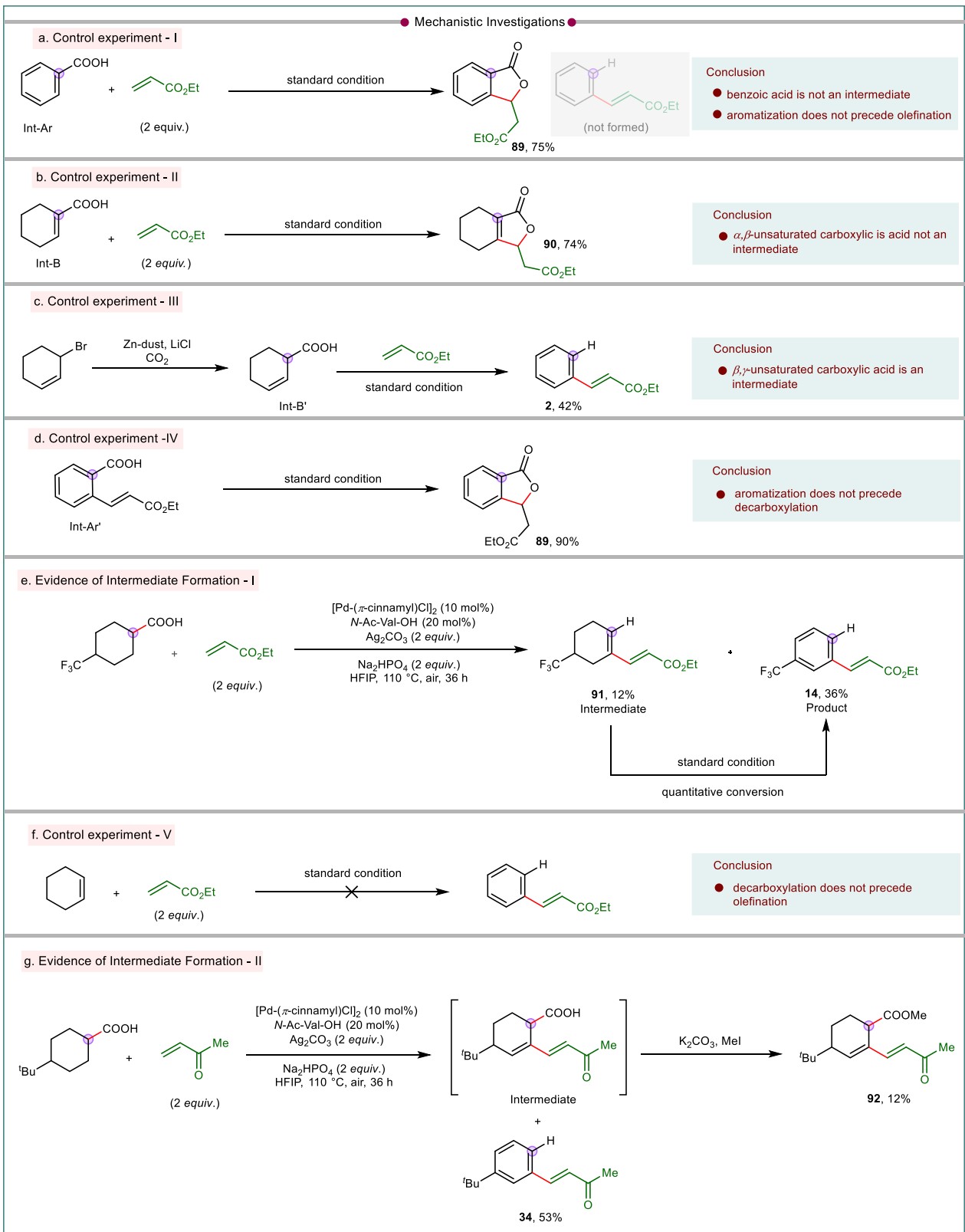

**Fig. 7 | Mechanistic insights.** Control experiments and isolation of probable intermediates. **a** Control experiment-I. **b** Control experiment-II. **c** Control experiment-III. **d** Control experiment-IV. **e** Evidence of intermediate formation-I. **f** Control experiment-V. **g** Evidence of intermediate formation-II. Standard condition: acid (0.1 mmol), olefin (2 *equiv.*), [Pd(π-cinnamyl)Cl]$_2$ (10 mol%), *N*-Ac-Val-OH (20 mol%), Ag$_2$CO$_3$ (2 *equiv.*), Na$_2$HPO$_4$ (2 *equiv.*), HFIP (1 mL), 110 °C, air, 36 h.

furnished the desired olefinated benzene **2** in 42% yield (Fig. 7c). This outcome strongly indicates the involvement of Int-B' in the proposed reaction pathway and olefination precedes over aromatization. For the preference of *β,γ*-dehydrogenated intermediate formation over

*α,β*-dehydrogenation, we reasoned that the dehydrogenation site (*β*-hydride elimination site) is controlled by the catalytic system, i.e., catalyst and the ligand (see Supplementary Information, Section 3.1.d for detailed analysis)[34,47,70]. Further, to gain insights about the order in

which decarboxylation and aromatization occurs, the present reaction condition was executed on ortho-olefinated benzoic acid (Int-Ar') (Fig. 7d). The formation of benzolactone **89** instead of the anticipated decarboxylated olefinated product **2** rules out the possibility of aromatization prior to decarboxylation. Based on the cumulative control experiments conducted, the following inferences can be drawn: (a) the reaction pathway involves the intermediacy of $\beta,\gamma$-unsaturated cyclohexene carboxylic acid (Int-B'); (b) olefination takes place prior to the aromatization step; (c) decarboxylation does not occur subsequent to the aromatization step and (d) aromatization is likely the final phenomenon in the entire reaction sequence (for details of the control reactions and the mechanistic studies please see Supplementary Information, Section 3.1. Detailed mechanistic investigation). Analogous mechanistic sequences involving C−H functionalization and decarboxylative aromatization also had been identified in the $C(sp)^2$−H activation of cyclohexa-2,5-diene-1-carboxylic acid, providing further support for the current hypothesis[71–73].

Worth-mentioning, when the reaction of 4-trifluoromethyl substituted cyclohexane carboxylic acid was intercepted after 20 h reaction time, along with the desired olefinated arene **14**, the formation of a decarboxylated-olefinated-cyclohexene **91**, in 12% yield was observed (Fig. 7e). Traces of the similar intermediates were also observed for 4-ethyl substituted and thymol tethered cyclohexane carboxylic acids (see Supplementary Information, Section 3.1.b). Expecting the decarboxylated-olefinated-cyclohexene (**91**) to be an intermediate, we tested the side product under our standard reaction protocol and to our delight, we found our desired product **14** formation in quantitative yield; confirming involvement of **91** as an intermediate of the reaction protocol. However, the intermediacy of **91** raises a query about the sequence of event between (a) Pd-catalyzed olefination at the $\beta−C(sp^2)$−H center, followed by decarboxylation or (b) decarboxylation followed by $C(sp^2)$−H olefination. The occurrence of the latter pathway would lead to cyclohexene which is expected to undergo further olefination.

However, the non-compatibility of cyclohexene to provide either the olefinated cyclohexene or the olefinated arene **2** eliminates the opportunity of the latter pathway (Fig. 7f). Also the formation of regiospecific olefinated arenes at the $\beta$- position with respect to carboxylic acid in cases of substituted cyclohexane carboxylic acid clearly indicates that olefination is driven by the directing assistance of carboxylic acid. Noteworthy, the low yield observed for the formation of arene products without the olefin coupling partner suggests that the olefination accelerates the decarboxylative aromatization. Hence the intermediacy of olefinated-cyclohexene **91** observed for 4-trifluoromethyl substituted cyclohexane carboxylic acid is occurring via decarboxylation induced double bond migration of the initially formed $\beta,\gamma$-unsaturated olefinated cyclohexyl carboxylic acid. Indeed, such an intermediate ($\beta,\gamma$-unsaturated olefinated cyclohexyl carboxylic acid, **92**, Fig. 7g) could be intercepted in case of the reaction of 4-'Bu cyclohexyl carboxylic acid with methyl vinyl ketone, further strengthening our hypothesis for the mechanistic cycle. From the observations of control experiments and the intercepted intermediates, it is evident that the sequence of reaction follows dehydrogenation-olefination-decarboxylation-aromatization pathways. A plausible mechanistic cycle is proposed based on D-O-D-A sequence (Fig. 8a: Plausible mechanisms for D-O-D-A). The acid substrate initially binds with the catalyst with the help of bidentate ligand. Next, the alkali metal $Na^+$ displaces Pd(II) from $\kappa_2$ coordination to $\kappa_1$ coordination and assists in C−H activation to generate Int-A. A $\beta$-hydride elimination from palladacycle Int-A delivers $\beta,\gamma$-dehydrogenated Int-B'. After the regeneration of the Pd(II) species by the oxidant (i.e., aerial oxygen and $Ag_2CO_3$), acid directed $C(sp^2)$−H activation provides vinyl palladium species Int-C'. This intermediate subsequently couples with olefin through olefin coordination, migratory insertion and $\beta$-hydride elimination to generate Int-D'. Int-D' undergo subsequent proto-decarboxylation[74] with an

olefinic shift to form Int-E. Thereafter Int-E, in presence of Pd(II), undergoes two consecutive allylic $C(sp^3)$−H activations/$\beta$-hydride elimination steps to deliver the olefinated benzene derivatives. The re-oxidation of Pd(0) to Pd(II) by the oxidant system carry forwards the next catalytic cycle. While for the five-membered cyclic carboxylic acids, the initial dehydrogenation-olefination-decarboxylation occurs as demonstrated for six-membered carboxylic acids. The olefinated cyclopentene then undergoes an allylic $C(sp^3)$−H activation to $\pi$-allyl palladium species (Fig. 8b: Plausible mechanisms for D-O-D-AA). At this stage instead of undergoing a further $\beta$-hydride elimination to give the diene product, the catalytic system controls the dehydrogenation and follows an alternative allylic acyloxylation[56,75,76] with the acid (either self-coupling or cross-coupling) to afford the difunctionalized cyclopentene derivatives (see Supplementary Information, Section 3.1.e for the D-O-D-AA detailed mechanism).

In summary, we have devised a catalytic system which utilizes multifold C−H activation to afford olefinated arenes starting from cyclohexyl carboxylic acids via sequential dehydrogenation-olefination-decarboxylation-aromatization. This protocol demonstrates the variation in reactivity of cyclohexane carboxylic acid with a subtle variation in their structural pattern. While reactions of aliphatic carboxylic acids with olefins are more known for lactone formation, in the present case the lactone formation could be eliminated by passing over to decarboxylation pathway. A number of acrylates, acryl amides, vinyl phosphonate, diesters, acrylonitrile, vinyl sulfones, unsaturated ketone have been successfully utilized in the reaction. Unactivated and semi-activated olefins, such as styrenes and allyl alcohols were also found to be compatible for our reaction protocol. Very unusual reactivity was observed for cyclohexane carboxylic acids possessing hydroxy substituent which underwent dehydroxylation instead of dehydrogenation step. Cycloalkyl carboxylic acids for which aromatic stabilization is not possible, formed difunctionalized cycloalkenes by controlled dehydrogenation following dehydrogenation-olefination-decarboxylation-allylic acyloxylation reaction sequence. The intricacies of the transformation were resolved by thorough control experiments and isolation of intermediates. These findings provide valuable insights into the mechanistic pathway and the sequential order of the key transformations involved. The protocol could also be applied for the synthesis of late-stage functionalized drugs or important precursors for synthesis of various bio-relevant molecules. Such an exhibition of reactivity changes in acid substrates can be harnessed to create a diverse array of applications.

## Methods

### General procedure for the synthesis of olefinated arenes

A clean, oven-dried screw cap reaction tube with previously placed magnetic stir−bar was charged with aliphatic cyclohexane carboxylic acid (0.1 mmol, 1 *equiv.*), olefinic partner (0.2 mmol, 2 *equiv.*), [Pd($\pi$-cinnamyl)Cl]$_2$ (0.01 mmol, 10 mol%), *N*-Ac-Val-OH (0.02 mmol, 20 mol%), $Ag_2CO_3$ (0.2 mmol, 2 *equiv.*) and $Na_2HPO_4$ (0.2 mmol, 2 *equiv.*) followed by addition of HFIP (1 mL) under air. The reaction mixture was vigorously stirred for 36 h in a preheated oil bath at 110 °C. After stipulated time, the reaction mixture was cooled to room temperature and filtered through a celite bed using ethyl acetate as the eluent (15 mL). The diluted ethyl acetate solution of the reaction mixture was subsequently washed with saturated brine solution (2 × 10 mL) followed by water (2 × 10 mL). The ethyl acetate layer was dried over anhydrous $Na_2SO_4$ and the volatiles were removed under vacuum. The crude reaction mixture was purified by column chromatography using silica gel and petroleum-ether /ethyl acetate as the eluent to give the desired olefinated-arene as the product.

### General procedure for the synthesis of arenes (without olefin)

A clean, oven-dried screw cap reaction tube with previously placed magnetic stir−bar was charged with aliphatic cyclohexane carboxylic

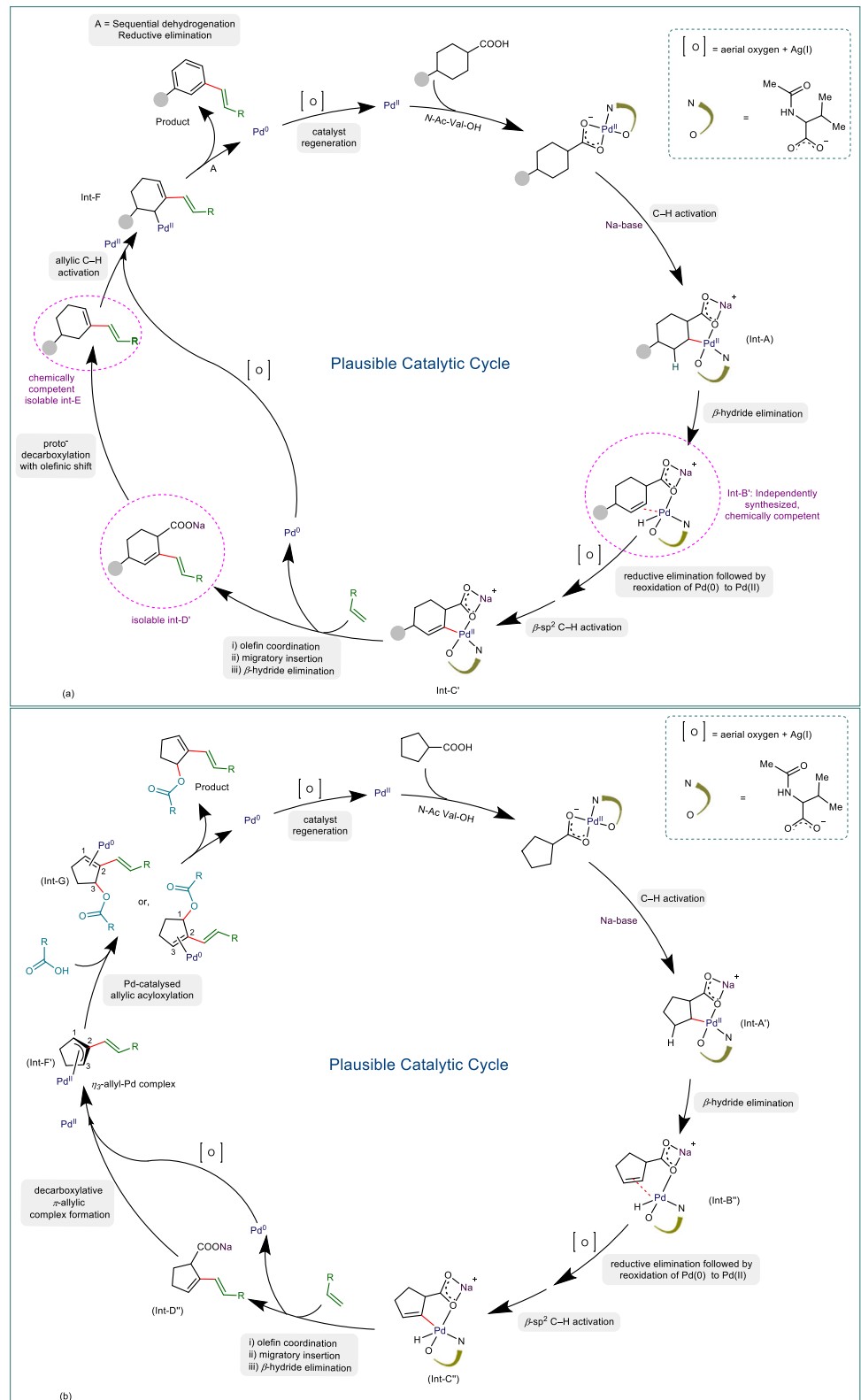

**Fig. 8 | Plausible mechanisms for D-O-D-A and D-O-D-AA. a** Plausible mechanism of dehydrogenation-olefination-decarboxylation-aromatization for cyclohexane carboxylic acid. **b** Plausible mechanism of dehydrogenation-olefination-decarboxylation-allylic acyloxylation for cyclopentane carboxylic acid (aerial oxygen: attaining oxygen present in the air).

acid (0.1 mmol, 1 *equiv*.), [Pd(*π*-cinnamyl)Cl]$_2$ (0.01 mmol, 10 mol%), *N*-Ac-Val-OH (0.02 mmol, 20 mol%), Ag$_2$CO$_3$ (0.2 mmol, 2 *equiv*.) and Na$_2$HPO$_4$ (0.2 mmol, 2 *equiv*.) followed by addition of HFIP (1 mL) under air. The reaction mixture was vigorously stirred for 36 h in a

preheated oil bath at 110 °C. After stipulated time, the reaction mixture was cooled to room temperature and filtered through a celite bed using ethyl acetate as the eluent (15 mL). The diluted ethyl acetate solution of the reaction mixture was subsequently

washed with saturated brine solution (2 × 10 mL) followed by water (2 × 10 mL). The ethyl acetate layer was dried over anhydrous $Na_2SO_4$ and the volatiles were removed under vacuum. The crude reaction mixture was purified by column chromatography using silica gel and petroleum-ether/ethyl acetate as the eluent to give the desired arenes as the product.

**General procedure for the multicomponent difunctionalization reaction**

A clean, oven-dried screw cap reaction tube with previously placed magnetic stir–bar was charged with cyclopentane carboxylic acid (0.1 mmol, 1 *equiv.*), cycloalkyl carboxylic acid (4 to 6 membered cycloalkyl carboxylic acid) (0.1 mmol, 1 *equiv.*), olefinic partner (0.2 mmol, 2 *equiv.*), [Pd($\pi$-cinnamyl)Cl]$_2$ (0.01 mmol, 10 mol%), *N*-Ac-Val-OH (0.02 mmol, 20 mol%), $Ag_2CO_3$ (0.2 mmol, 2 *equiv.*) and $Na_2HPO_4$ (0.2 mmol, 2 *equiv.*) followed by addition of HFIP (1 mL) under air. The reaction mixture was vigorously stirred for 36 h in a preheated oil bath at 110 °C. After stipulated time, the reaction mixture was cooled to room temperature and filtered through a celite bed using ethyl acetate as the eluent (15 mL). The diluted ethyl acetate solution of the reaction mixture was subsequently washed with saturated brine solution (2 × 10 mL) followed by water (2 × 10 mL). The ethyl acetate layer was dried over anhydrous $Na_2SO_4$ and the volatiles were removed under vacuum. The crude reaction mixture was purified by column chromatography using silica gel and petroleum-ether / ethyl acetate as the eluent to give the desired difunctionalized cyclopentene derivative.

## Data availability

All data supporting the findings of this study including experimental procedures and compound (starting materials, products) characterization are available in the Supplementary Information.

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

## Acknowledgements
Financial support received from SERB CRG (CRG/2022/004197) and IoE-IITB (IoE WBS: IE/21-ACADEMIC-IF-A2_GEN) are gratefully acknowledged. Financial support received as fellowship from PMRF-India (T.P., M.I., and S.D.), IITB-Monash Research Academy (P.G.) and CSIR-India (S.M. and J.D.) are gratefully acknowledged. H.G. would like to acknowledge NSF (CHE-2029932), Robert A. Welch Foundation (D-2034-20230405), and Texas Tech University for financial support.

## Author contributions
D.M. and T.P. conceived the concept. T.P., P.G., M.I., S.G., S.M., and S.D. performed the reactions and analyzed the products. T.P., S.G., J.D., H.G., and D.M. wrote the manuscript.

## Competing interests
The authors declare no competing interests.
