## [Peer Review File · Nature Communications]

REVIEWER COMMENTS

Reviewer #1 (Remarks to the Author):

Comments: The authors have presented a novel method involving a tandem dehydrogenation-olefination-decarboxylation of cycloalkyl carboxylic acids through multiple C–H activation steps. This innovative approach yields intriguing results, and the authors have also given well explanation strategies of organic molecules formation (D-O-D-A, D-O-D-D-A, O-D-D-A and D-O-D-AA) through detailed mechanistic investigation, Int-B' (control experiment - III) was key intermediate for this transformation. Although the manuscript contains several minor grammatical errors that do not impact the readability of the text. Overall the manuscript is suitable for publication after major revisions are noted.

Comment 1. This reviewer suggests that an important related work from Prof. Jin-Quan Yu need to be cited (J. Am. Chem. Soc. 2023, 145, 38, 20951–20958).

Comment 2. The author should provide an explanation for why allylated arene derivatives (compounds 30-31) are formed instead of vinylated arene products.

Comment 3. Scheme 7c and 7g necessitate a more comprehensive background explanation regarding the C-H olefinations from β , γ -unsaturated cyclohexene carboxylic acid. These synthesis methods were previously explored by Chou's group (Org. Lett. 2018, 20, 5, 1328–1332 and Org. Lett. 2020, 22, 17, 6765–6770 and Arkivoc 2023 (ii) 202312004).

Comment 4. In comparison to the final product, Intermediates D and E appear to be promising substrates due to their possession of conjugated dienes with tethered carboxylates. The author should investigate the feasibility of controlling the reaction at the Intermediates D and E stages by adjusting the choice of oxidizing agent in order to isolate these intermediates in high yields (Org. Lett. 2020, 22, 17, 6765–6770).

Comment 5. How about using the substrate of 1-substituted cyclohexyl carboxylic acid? Additional substituent attached on that position may perturb the rate of decarboxylation /aromatization.

Comment 6. The author proposed a Pd/Ag catalytic cycle. It seems Ag(I) served as an oxidant for three time in the proposed catalytic cycle, but only 2 equivalents of Ag₂CO₃ was used as a standard condition. Any explanation?

Comment 7. The author is encouraged to give a detail description of reaction mechanism of Scheme 5.

Comment 8. The weight and amount of reagents/products should be reported

Reviewer #2 (Remarks to the Author):

Maiti and co-workers present a tandem multifold C–H activation process of cycloalkyl carboxylic acids, integrating dehydrogenation, olefination, and decarboxylation in the same catalytic system. This represents a significant advancement in the reactivity of aliphatic carboxylic acids, demonstrating good substrate applicability for constructing complex molecular skeletons and synthesizing drug precursors. The manuscript is well-presented with clear and concise data in both the main text and supporting information. The authors conduct detailed mechanistic studies, revealing that the potential reaction intermediate is β , γ -unsaturated carboxylic acid rather than benzoic acid or α , β -unsaturated carboxylic acid. Considering these strengths, the manuscript is deemed worthy of publication in Nature Communications after addressing minor revisions.

1. In the optimization of reaction conditions and catalyst screening (Table S1), different Pd precursors exhibit varying reactivity and chemo-selectivity, notably [Pd(π -cinnamyl)Cl]₂, which inhibits bis-dehydrogenated lactone byproduct. Given that the catalytic Pd species should be the same in the catalytic cycle, could you explain the reasons behind the high chemo-selectivity with [Pd(π -cinnamyl)Cl]₂?

2. The author mentions that a phenol derivative was expected, but the presence of a free -OH group could poison the catalyst and inhibit the reaction. In the DODDA reactions, dehydroxylation occurs between decarboxylation and aromatization. Do you have additional information or evidence supporting this dehydroxylation?

3. If the -OH group is protected in the DODDA reactions, what would happen? Additionally, if there is a -OH group in the 5-membered aliphatic cyclic acids for the difunctionalized cyclopentene derivatives, is dehydroxylation still possible?

4. In many Ag₂CO₃-involved C–H activation reactions, Ag₂CO₃ serves as both an oxidant and a base. How does it act in the displayed reactions? If the base Na₂HPO₄ is removed, what would happen? Please add this data to the base optimization table in the SI.

5. HFIP is an outstanding solvent in many C–H activation reactions. For most of the screened solvents in the SI, the results are listed as nr, but HFIP shows a fantastic improvement. Could you provide a reasonable explanation for this?

6. The author conducts mechanistic experiments, figuring out the possible intermediate. Can you explain why decarboxylation happens on int-D', rather than int-C' or int-E/F?

7. Multiple C–H activation and β-H elimination occur in the plausible catalytic cycle. If a deuterated carboxylic acid or deuterated solvent (D₂-HFIP) were used, would there be any H-D exchange or D-incorporation in the final product?

Reviewer #3 (Remarks to the Author):

The authors achieved tandem oxidative transformation of cyclohexyl and cyclopentyl carboxylic acids via Pd-catalyzed multi-C-H activation. The reaction with cyclohexyl substrates provided aromatized compounds, while cyclopentyl substrates gave allylic oxidation adducts. The mechanistic studies described in Scheme 7 are reasonable, supporting the tandem sequence described in postulated catalytic cycle.

On the other hand, this work can be regarded as substrate-controlled reactions, rather than catalyst-controlled one, because similar reaction conditions gave different lactone adducts from somewhat modified starting materials. Thus, novelty on the catalyst is limited and the value of the manuscript should be evaluated based on its synthetic novelty and utility. As to the synthetic novelty, the present protocol promoted various different reactions, i.e.

dehydrogenation/alkenylation/decarboxylation/aromatization, in one-pot. That is nice. Synthetic utility of the work is, however, somewhat limited due to limited availability of the starting materials. The present method provided meta-di-substituted aromatic compounds, which is useful building blocks. But, compared with other available methods, like Heck reaction of aryl halides, diversity of products is not sufficient. In case of previous reports of phenol synthesis from cyclohexanol, the diversity of products is well documented, while in this work the authors just utilized substrates with limited structural diversity in Scheme 2. The use of styrenes in Scheme 3 is nice, but the scope was limited to electron-deficient substrates. If the dehydration can be suppressed in Scheme 4a to access phenol derivatives, the synthetic utility might increase significantly. But, unfortunately, hydroxy-substituted acids gave decarboxylation/dehydration products. Demonstration of synthetic utility in Scheme 6 is not satisfactory as the protocol only provided relatively simple synthetic intermediates, which can be supplied in a different manner.

Thus, I am afraid the manuscript does not reach to the level required for publication in Nat Commun. Re-submission to other journals specialized on chemistry after major revisions to clarify several unclear points are recommended.

1) The amount of oxidant: The authors utilized only two equiv of oxidant. But, in the proposed catalytic cycle, re-oxidation of Pd(0) to Pd(II) is required more than twice, including quenching Pd-hydride species. Did authors perform the reaction under air/O₂ or inert gas? If the reaction was performed under inert gas, excess alkene might work as hydride acceptor. Please carefully re-consider the regeneration of Pd active species from Pd-hydride and include it in the catalytic cycle. The yield of products decreased in the absence of alkenes (Scheme4b), supporting the role of excess alkenes as Pd-hydride acceptor.

2) Scheme 5: The authors indicated that the oxidation proceeded site-selectively by showing decarboxylation site and acid addition site separately. I am afraid that is clearly mis-leading readers. If the reaction proceeds via symmetric π -allyl Pd, then, both sites should be oxidized. Thus, it is impossible to differentiate decarboxylation site and acid addition site.

3) In SI, information on starting materials may be missing. Did authors use only commercially available carboxylic acids as described in page 3?

4) Dr of products 10 and 11 are not reported.

Response Letter

Reviewer 1

The authors have presented a novel method involving a tandem dehydrogenation-olefination-decarboxylation of cycloalkyl carboxylic acids through multiple C–H activation steps. This innovative approach yields intriguing results, and the authors have also given well explanation strategies of organic molecules formation (D-O-D-A, D-O-D-D-A, O-D-D-A and D-O-D-AA) through detailed mechanistic investigation, Int-B' (control experiment - III) was key intermediate for this transformation. Although the manuscript contains several minor grammatical errors that do not impact the readability of the text. Overall, the manuscript is suitable for publication after major revisions are noted.

We are thankful to the reviewer for favorable recommendation and also for the constructive suggestions to improve the manuscript and supporting information. We have addressed the queries and necessary modifications have been included in the main manuscript and supporting information.

Reviewer's comment 1: This reviewer suggests that an important related work from Prof. Jin-Quan Yu need to be cited (J. Am. Chem. Soc. 2023, 145, 38, 20951–20958).

Our response: As per reviewer's suggestion, we cited the relevant work as reference number **70** in the revised manuscript and highlighted in yellow.

Reviewer's comment 2: The author should provide an explanation for why allylated arene derivatives (compounds 30-31) are formed instead of vinylated arene products. (compounds 30-31 now have been renumbered as compounds 32-33).

Our response: We thank the reviewer for insightful comment. The rationale behind the formation of allylated products instead of vinylated products in the cases of cyclic tri-substituted olefins (previously listed as entry **30** & **31**, now as **32** & **33** in the revised manuscript) can be attributed to steric factors present in the intermediates. Pd-catalyzed C–H olefination reaction involves β -hydride elimination, where β -hydride should be in *syn*-periplanar to the metal center (Pd). In the intermediate involving a cyclic tri-substituted olefins, *syn*- β -hydride elimination is only facile from a site that produces allylated product.

It is highly likely that the red β -hydride is not in a *syn*-periplanar geometry with the Pd-catalyst, while the blue β -hydride achieves *syn* coplanarity, leading to the generation of allylated arenes. Allylated products frequently form when cycloalkenes are used, often surpassing the formation of vinylated products. This trend, including allylic shifts, is well-documented in the literature, not only in C–H activation reactions (Patra *et al.*, *Chem. Commun.* **52**, 2027-2030 (2016)), but also in Heck coupling reactions (Crisp *et al.*,

Chem. Soc. Rev. **27**, 427-436 (1998)). In the revised manuscript, these references have been cited as reference numbers **57** and **58** respectively, with the following line:

“The occurrence of such allylation can be explained by the fact that, in the intermediate containing cyclic tri-substituted olefins, *syn*- β -hydride elimination is only facile from a site favoring the production of allylated products.”

Reviewer’s comment 3: Scheme 7c and 7g necessitate a more comprehensive background explanation regarding the C–H olefinations from β,γ -unsaturated cyclohexene carboxylic acid. These synthesis methods were previously explored by Chou’s group (*Org. Lett.* 2018, 20, 5, 1328–1332 and *Org. Lett.* 2020, 22, 17, 6765–6770 and *Arkivoc* 2023 (ii) 202312004).

Our response: We thank the reviewer for valuable suggestion.

To obtain a clear vision on the preference for β,γ -unsaturation than the α,β -unsaturation, we reasoned that the catalytic system (Pd-catalyst and the ligand involved) controlled site-selective β -hydride elimination. To investigate the site selectivity in dehydrogenation, we performed a couple of reactions with 4-(*tert*-butyl)cyclohexane carboxylic acid using quinoline-pyridone ligand (L_D) instead of *N*-Ac-Val-OH, keeping reaction parameters same as that of our optimized conditions.

It was observed that in absence of olefin coupling partner, the substrate provided 4-(*tert*-butyl)cyclohex-1-ene-1-carboxylic acid (i.e. having α,β -unsaturation) in 38% yield. While in presence of ethyl acrylate, the reaction provided 30% of the α,β -unsaturated acid along with 8% formation of doubly dehydrogenated γ -lactone derivative.

Also from the literature report, it is evident that the ligand, together with the catalyst, governs selective β -hydride elimination (a. Sheng *et al.*, *J. Am. Chem. Soc.* **144**, 12924-12933 (2022), b. Wang *et al.*, *Science*. **374**, 1281-1285 (2021), c. Sheng *et al.*, *J. Am. Chem. Soc.* **145**, 20951–20958 (2023)). This further concludes the catalytic system enabled site selective dehydrogenation of aliphatic carboxylic acid. We have discussed these observations in details in the SI section 8. (d) (page no.77-78). Additionally, in the revised main manuscript, we have added a line with the aforementioned references, stated as: “For the preference of β,γ -dehydrogenated intermediate formation over α,β -dehydrogenation, we reasoned that

the dehydrogenation site (β -hydride elimination site) is controlled by the catalytic system, i.e. catalyst and the ligand (see Supporting Information, Section 7.d for detailed analysis).^{70,34,47}”

Furthermore, in support of our mechanistic observations in C–H olefinations from β,γ -unsaturated cyclohexene carboxylic acid, we have cited the following publications in the revised manuscript: a) Tsai *et al.*, *Org. Lett.* **20**, 1328–1332 (2018). b) Wang *et al.*, *Org. Lett.* **22**, 6765–6770 (2020). c) Kadiyala *et al.*, *Arkivoc* (**ii**) 202312004 (2023) and we have also incorporated the following sentence in the revised manuscript: “Analogous mechanistic sequences involving C–H functionalization and decarboxylative aromatization also had been identified in the $C(sp)^2$ –H activation of cyclohexa-2,5-diene-1-carboxylic acid, providing further support for the current hypothesis.⁷¹⁻⁷³”

Reviewer’s comment 4: In comparison to the final product, Intermediates D and E appear to be promising substrates due to their possession of conjugated dienes with tethered carboxylates. The author should investigate the feasibility of controlling the reaction at the Intermediates D and E stages by adjusting the choice of oxidizing agent in order to isolate these intermediates in high yields (*Org. Lett.* 2020, 22, 17, 6765–6770).

Our response: We thank the reviewer for this constructive suggestion.

Since we were able to obtain intermediate D with methyl vinyl ketone in a shorter reaction time using our standard reaction parameters, we proceeded to optimize oxidants using 4-*t*Bu-cyclohexyl carboxylic acid and MVK (methyl vinyl ketone) as reactants to achieve intermediate **D** as the sole product. We also tried to intercept such diene intermediate (**Int-D**) with an olefin (*N*-methyl maleimide) under standard conditions. However, the yield of the mono-unsaturated acid (Int-D) could not be maximized with the different types of oxidants used. The optimization table is provided below:

Entry	Oxidant	NMR Yield (A) (%)	NMR Yield (B) (%)
1	Ag ₂ CO ₃ with N -methyl maleimide	N.R.	N.R.
2	AgOAc	10%	0%

3	AgI	N.R.	N.R.
4	AgBr	N.R.	N.R.
5	AgTFA	N.R.	N.R.
6	Ag ₃ PO ₄	Trace	0%
7	Cu(OAc) ₂	Trace	0%
8	Ag ₂ CO ₃ + Benzoquinone (2:1)	34%	8%
9	Ag ₂ CO ₃ + K ₂ S ₂ O ₈ (2:1)	31%	6%
10	CF ₃ SO ₃ Ag	38%	0%
11	AgSbF ₆	N.R.	N.R.

Remarkably, for the reaction of 4-methoxy-substituted cyclohexane carboxylic acid with ethyl acrylate under the standard condition, we observed olefination-decarboxylation, accompanied by controlled dehydrogenation, resulting in an olefinated cyclohexene derivative, which was eventually identified as the same intermediate **E**. This experiment led us to the conclusion that decarboxylative controlled dehydrogenation can likely be achieved through substrate-controlled C–H functionalization. We have incorporated this data in the section 10 of the revised supplementary information (page no. 82).

Reviewer's comment 5: How about using the substrate of 1-substituted cyclohexyl carboxylic acid? Additional substituent attached on that position may perturb the rate of decarboxylation / aromatization.

Our response: We appreciate the insightful suggestion from the reviewer. Following the recommendation, we have conducted experiments with two substrates containing 1-alkyl substituted cyclohexyl carboxylic acid, successfully expanding the scope of our protocol. These additional scopes have been incorporated as **23** & **24** in Scheme 2 of the revised manuscript. The characterization data of these synthesized products have been included in the revised SI (page 33-35). Furthermore, a dedicated paragraph addressing this different substrate class has been also included in the manuscript.

“Next, we sought to examine 1-substituted cyclohexane carboxylic acids. Despite having an α -quarternary centre, the decarboxylation anyway facilitated the aromatization process in such acid classes. It has been observed that 1-methyl and 1-propyl substituted cyclohexane carboxylic acids

underwent to D-O-D-A sequence yielding 2-alkyl substituted olefinated benzene (23-24). However, minor amount of gamma-lactones were also formed as side products in these cases (see Supporting Information, Entries 23' and 24').”

Reviewer's comment 6: The author proposed a Pd/Ag catalytic cycle. It seems Ag(I) served as an oxidant for three time in the proposed catalytic cycle, but only 2 equivalents of Ag_2CO_3 was used as a standard condition. Any explanation?

Our response: We thank the reviewer for this constructive evaluation. In the proposed catalytic system, re-oxidation of Pd(0) to Pd(II) happened more than thrice. However, it was observed that 2 equivalents of Ag(I) were optimal for this reaction. To delve deeper into this phenomenon, we conducted several controlled experiments, as outlined below:

*Entry	Oxidant	NMR Yield (%)
1	Ag_2CO_3	61
2	4.0 equiv. Ag_2CO_3	58
3	Ag_2CO_3 under argon atmosphere	trace
4	Ag_2CO_3 under nitrogen atmosphere	28%

5	4.0 equiv. Ag ₂ CO ₃ under nitrogen atmosphere	44%
---	---	-----

(*It is only a comparison table using different environment. For a detailed and full oxidant optimization studies, please see, Supporting Information, **Table S7** and **Table S8**, page no. 8-10).

From the oxidant optimizations, it is evident that Ag(I) salt alone is not effective as an oxidant in this particular transformation. Notably, experiments conducted under inert atmosphere revealed that even a super stoichiometric amount of silver carbonate failed to provide similar yield (Entry 3, 4, 5) compared to the reaction using 2 *equiv.* of Ag₂CO₃ under air (Entry 2). Hence, it is reasonable to conclude that the synergistic action of Ag(I) and aerial oxygen serves as an oxidant system in this reaction. Such observation is also supported by the literature (Izawa *et al. Science*, **333**, 209-213 (2011)), where dehydrogenation / cyclic ring aromatization was facilitated using Pd-catalyst in combination with O₂ oxidant. Accordingly, we have specified ‘air’ in each of the reaction schemes as an oxidant system along with Ag₂CO₃ in the MS and SI. Additionally, detailed discussions regarding the role of atmospheric oxygen have been provided in the supplementary information section 2 (page no. 10).

Reviewer’s comment 7: The author is encouraged to give a detail description of reaction mechanism of Scheme 5.

Our response: We express our gratitude to the reviewer for their meticulous evaluation. The plausible mechanism for Scheme 5, which was previously presented in the supplementary information, has now been incorporated into the main manuscript as Scheme 8 (b) with slight modifications and highlighted in yellow. The mechanism alters the likelihood of consecutive dehydrogenation by introducing an alternative allylic C–H acyloxylation reaction. A detailed description of this mechanism has been included in the supporting information (Section 8. e., page no. 78-79).

Reviewer's comment 8: The weight and amount of reagents/products should be reported.

Our response: We appreciate the valuable suggestion from the reviewer. In the revised Supplementary Information (SI) and main manuscript (MS), we have now included the weight and amount of reagents/products.

Reviewer 2

Maiti and co-workers present a tandem multifold C–H activation process of cycloalkyl carboxylic acids, integrating dehydrogenation, olefination, and decarboxylation in the same catalytic system. This represents a significant advancement in the reactivity of aliphatic carboxylic acids, demonstrating good substrate applicability for constructing complex molecular skeletons and synthesizing drug precursors. The manuscript is well-presented with clear and concise data in both the main text and supporting information. The authors conduct detailed mechanistic studies, revealing that the potential reaction intermediate is β , γ -unsaturated carboxylic acid rather than benzoic acid or α , β -unsaturated carboxylic acid. Considering these strengths, the manuscript is deemed worthy of publication in *Nature Communications* after addressing minor revisions.

We are extremely grateful to the reviewer for the kind appreciation and supporting the publication of this work. We have included the modifications in the revised manuscript and supporting information addressing the reviewers' comments and suggestions.

Reviewer's comment 1: In the optimization of reaction conditions and catalyst screening (Table S1), different Pd precursors exhibit varying reactivity and chemo-selectivity, notably [Pd(π -cinnamyl)Cl]₂, which inhibits bis-dehydrogenated lactone byproduct. Given that the catalytic Pd species should be the same in the catalytic cycle, could you explain the reasons behind the high chemo-selectivity with [Pd(π -cinnamyl)Cl]₂?

Our response: We sincerely appreciate the reviewer's thoughtful comment. Experimentally, we are unable to conclude the exact role of [Pd(π -cinnamyl)Cl]₂ for the chemo-selectivity. However, from the Pd-catalyst optimizations (catalyst optimization table is given below, already in the SI as Table S1, section 2), it can be hypothesized that some sort of bulkiness and the π -assistance from counter anionic parts of the Pd-catalyst may influence the reaction outcome. The π -allyl Pd complex also provided the olefinated arene as the sole product. Currently, our laboratory is actively conducting detailed mechanistic and computational (DFT) studies to gain further insights.

Entry	Catalyst	NMR Yield (1) (%)	Side Product (1') (%)
1	Pd(OAc) ₂	38	16
2	Pd(PPh ₃) ₂ Cl ₂	28	12
3	Pd(PhCN) ₂ Cl ₂	20	4
4	[Pd(allyl)Cl]₂	11	0
5	[Pd(π-cinnamyl)Cl]₂	53	0
6	Pd(OPiv) ₂	20	16
7	Pd(COD)Cl ₂	53	12
8	PdCl ₂	10	trace
9	Pd(acac) ₂	7	trace
10	Pd(CH ₃ CN) ₂ Cl ₂	31	7
11	Pd ₂ (dba) ₃	58	7
12	Pd(dba) ₂	43	6
13	Pd(dppf) ₂ Cl ₂	19	5
14	Pd(MeCN) ₄ BF ₄	trace	trace
15	Pd/C	n.r.	n.r.

Reviewer's comment 2: The author mentions that a phenol derivative was expected, but the presence of a free -OH group could poison the catalyst and inhibit the reaction. In the DODDA reactions,

dehydroxylation occurs between decarboxylation and aromatization. Do you have additional information or evidence supporting this dehydroxylation?

Our response: We express our gratitude to the reviewer for the valuable comment. The mechanistic cycle involving hydroxy-substituted carboxylic acid underwent β -hydroxy elimination.

Metal catalyzed β -hydroxy elimination is a well-documented reaction in literature (a. Hacksell *et al.*, *Organometallics*, **2**, 772–775 (1983), b. Bras *et al.*, *Tetrahedron*, **68**, 10065–10113 (2012), c. Dutta *et al.*, *Chem*, **7**, 555–605 (2021)). In our reactions also, we observed such phenomenon. To substantiate our observations, we have now cited these mentioned publications as references **60**, **61**, **62** in the revised manuscript.

Reviewer's comment 3: If the –OH group is protected in the DODDA reactions, what would happen? Additionally, if there is a –OH group in the 5-membered aliphatic cyclic acids for the difunctionalized cyclopentene derivatives, is dehydroxylation still possible?

Our response: We are grateful to the reviewer for the fruitful suggestion.

In accordance with the reviewer's insightful suggestion, we synthesized 4-acetoxy-substituted cyclohexane carboxylic acid by the acyl protection of –OH group. With the synthesized acid under the established reaction protocol, we witnessed facile β -acetoxy elimination within the substrate resulting olefinated benzene. We have now included this entry in Scheme 4 and added a corresponding line in the revised manuscript as: “A comparable reactivity involving acetoxy elimination⁶¹⁻⁶² was observed when 4-acetoxy-substituted cyclohexyl carboxylic acid underwent olefination and decarboxylative aromatization, leading to the formation of ethyl cinnamate (2).”

However, when the hydroxyl group is protected with a methyl group, as in the case of 4-methoxy-substituted cyclohexyl carboxylic acid, it exhibited diminished reactivity towards aromatization under standard conditions with ethyl acrylate. Instead, it predominantly yielded a product with decarboxylation

and mono-dehydrogenation. We have incorporated this information in section 10: ‘Unsuccessful substrate’ (Entry **93**) of the revised supplementary information (page no. 82).

Introduction of a ‘–OH’ group in the 5-membered aliphatic cyclic acids did not seem to be fruitful in attaining any expected difunctionalized product.

The reaction with 3-hydroxy substituted cyclopentane carboxylic acid and ethyl acrylate, neither involved C–H activation nor olefination/decarboxylation; instead, an intramolecular lactonization reaction took place. We have included this entry as ‘Unsuccessful substrate’ (Entry **94**) in the section 10 of the revised SI (page no. 82-83).

Reviewer’s comment 4: In many Ag_2CO_3 -involved C–H activation reactions, Ag_2CO_3 serves as both an oxidant and a base. How does it act in the displayed reactions? If the base Na_2HPO_4 is removed, what would happen? Please add this data to the base optimization table in the SI.

Our response: We agree with the reviewer that in many C–H activation reactions involving Ag_2CO_3 , it serves the dual role of both an oxidant and a base. In response to the reviewer's valuable suggestion, we conducted a reaction without using Na_2HPO_4 base. Additionally, we performed another reaction without base, using 4 equiv. of Ag_2CO_3 to get a clearer idea for the role of the base used.

Entry	Without addition of Na_2HPO_4	NMR Yield (%)
1	Without base	18
2	without base, with 4 equiv. Ag_2CO_3	22

We found that the reaction could occur without the addition of an external base in the reaction medium, but it resulted in a very low yield of the product. Furthermore, even with the addition of 4 equivalents of silver carbonate, not much improvement in yield was encountered. These data have been included in the revised Supplementary Information, Table S9 (page no. 10).

This observation leads to the conclusion that Ag_2CO_3 can indeed function as both a base and an oxidant in this reaction, although it appears to be less effective than Na_2HPO_4 . The optimum condition involves usage of 2 *equiv.* Na_2HPO_4 as base and 2 *equiv.* Ag_2CO_3 as oxidant to achieve highest yield of the product (Table S9, Entry 24 in the SI, page no. 11).

Reviewer's comment 5: HFIP is an outstanding solvent in many C–H activation reactions. For most of the screened solvents in the SI, the results are listed as nr, but HFIP shows a fantastic improvement. Could you provide a reasonable explanation for this?

Our response: We extend our appreciation to the reviewer for the valuable comment. Based on the solvent optimization studies, it is clear that HFIP stands out as the most effective solvent for this reaction. The primary reasons attributing to the exceptional performance of HFIP solvent in this C–H activation reaction are as follows: a) HFIP possesses polar characteristics and is a potent hydrogen bond donor. This unique property enables HFIP to effectively coordinate with the substrate, thereby enhancing the reactivity of the reaction protocols. Furthermore, this enhanced coordination capability aids in stabilizing transition states. b) HFIP lessens the pH of the media, thereby helping in solubilization of acidic substrates.

The updated SI now incorporates this discussion (page no. 12), supported with a citation: Motiwala *et al.*, *Chem. Rev.* **122**, 12544–12747 (2022).

Reviewer's comment 6: The author conducts mechanistic experiments, figuring out the possible intermediate. Can you explain why decarboxylation happens on int-D', rather than int-C' or int-E/F?

Our response: We sincerely appreciate the insightful comments provided by the reviewer.

If decarboxylation were to proceed directly from cyclohex-2-ene-1-carboxylic acid, it would result in a cyclohexene lacking any functional group to further direct C–H functionalization in the unactivated alkene system. Hence, this pathway cannot be regarded as viable for decarboxylation. Additionally, we

conducted a controlled experiment (already documented in the manuscript, Scheme 7.f) involving cyclohexene and ethyl acrylate under standard conditions.

Hypothesis

Controlled experiment

The failure of this reaction confirms the validity of our hypothesized pathway and elucidates why decarboxylation does not occur from Int-B' or Int-C'.

However, it has been previously demonstrated through numerous controlled experiments, intermediate isolations, and mechanistic studies that Int-C', following Pd-catalyzed β -C(sp)²-H olefination, underwent proto-decarboxylation with an olefinic shift to provide Int-E. This intermediate (Int-E) subsequently proceeded towards aromatization (see Supporting Information, Section 8. Detailed mechanistic investigation). Consequently, this confirms that the decarboxylated Int-E can only be attained through the decarboxylation of Int-D' and not from Int-B' or Int-C'. Therefore, it is not feasible to achieve decarboxylation of an already decarboxylated substrate (Int-E/F).

Plausible mechanism

Moreover, we conducted tests using an α,β -unsaturated carboxylic acid to assess their potential contribution and feasibility in producing olefinated arenes. However, the observation of a dehydrogenated lactone derived from such an acid eliminated the possibility of an α,β -dehydrogenated acid as an intermediate in this reaction (as already detailed in the manuscript, Scheme 7.b) and hence, decarboxylation from such an intermediate does not occur.

Reviewer's comment 7: Multiple C–H activation and β -H elimination occur in the plausible catalytic cycle. If a deuterated carboxylic acid or deuterated solvent (D_2 -HFIP) were used, would there be any H–D exchange or D-incorporation in the final product?

Our response: We express our gratitude to the reviewer for the valuable comment. Based on the valuable recommendations from the reviewer, we conducted the H/D exchange experiments utilizing 4-*tert*-butyl cyclohexane carboxylic acid in the presence of D_2 -HFIP solvent, both with and without the inclusion of ethyl acrylate as a coupling partner.

Nevertheless, no deuterium incorporation is observed in the substrate for both the reactions, indicating the irreversibility of the C–H activation steps involved.

Reviewer 3

The authors achieved tandem oxidative transformation of cyclohexyl and cyclopentyl carboxylic acids via Pd-catalyzed multi-C–H activation. The reaction with cyclohexyl substrates provided aromatized compounds, while cyclopentyl substrates gave allylic oxidation adducts. The mechanistic studies described in Scheme 7 are reasonable, supporting the tandem sequence described in postulated catalytic cycle.

On the other hand, this work can be regarded as substrate-controlled reactions, rather than catalyst-controlled one, because similar reaction conditions gave different lactone adducts from somewhat modified starting materials. Thus, novelty on the catalyst is limited and the value of the manuscript should be evaluated based on its synthetic novelty and utility. As to the synthetic novelty, the present protocol promoted various different reactions, i.e. dehydrogenation/alkenylation/decarboxylation/aromatization, in one-pot. That is nice. Synthetic utility of the work is, however, somewhat limited due to limited availability of the starting materials. The present method provided meta-di-substituted aromatic compounds, which is useful building blocks. But, compared with other available methods, like Heck reaction of aryl halides, diversity of products is not sufficient. In case of previous reports of phenol synthesis from cyclohexanol, the diversity of products is well documented, while in this work the authors just utilized substrates with limited structural diversity in Scheme 2. The use of styrenes in Scheme 3 is nice, but the scope was limited to electron-deficient substrates. If the dehydration can be suppressed in Scheme 4a to access phenol derivatives, the synthetic utility might increase significantly. But, unfortunately, hydroxy-substituted acids gave decarboxylation/dehydration products. Demonstration of synthetic utility in Scheme 6 is not satisfactory as the protocol only provided relatively simple synthetic intermediates, which can be supplied in a different manner.

Thus, I am afraid the manuscript does not reach to the level required for publication in *Nat Commun*. Re-submission to other journals specialized on chemistry after major revisions to clarify several unclear points are recommended.

We thank the reviewer for insightful comments. The Reviewers' remarks helped us immensely to improve the quality of the manuscript. We trust that the reviewer will now find this revised version suitable for publication.

Our response: Site-selective dehydrogenation in a carboxylic acid is controlled by the ligand system with the catalyst. Changing the catalytic system (Pd and ligand) would produce alternative dehydrogenation (a. This work by Maiti and coworkers, b. Sheng *et al.*, *J. Am. Chem. Soc.* 144, 12924–12933 (2022), b. Wang *et al.*, *Science*. 374, 1281–1285 (2021), c. Sheng *et al.*, *J. Am. Chem. Soc.* 145, 20951–20958 (2023)). Additionally, we have successfully demonstrated decarboxylative functionalization in C(sp³)–H activation of aliphatic acids for the very first time. In conclusion, we demonstrated that, the reaction is not substrate-controlled, rather the outcome is mainly dependent on the catalytic system (ligand and the catalyst). The current protocol also includes a wide array of substrate scopes, demonstrating the broad applicability and generality of the established reaction protocol. A diverse range of *ortho*-, *meta*-, and *para*-disubstituted arenes, as well as di-functionalized cyclopentene derivatives, can be readily accessed through this methodology.

Reviewer's comment 1: The amount of oxidant: The authors utilized only two *equiv.* of oxidant. But, in the proposed catalytic cycle, re-oxidation of Pd(0) to Pd(II) is required more than twice, including quenching Pd-hydride species. Did authors perform the reaction under air/O₂ or inert gas? If the reaction was performed under inert gas, excess alkene might work as hydride acceptor. Please carefully reconsider the regeneration of Pd active species from Pd-hydride and include it in the catalytic cycle. The yield of products decreased in the absence of alkenes (Scheme 4b), supporting the role of excess alkenes as Pd-hydride acceptor.

Our response: We thank the reviewer for this constructive suggestion. In the proposed catalytic system, re-oxidation of Pd(0) to Pd(II) happened more than twice. However, it was observed that 2

equivalents of Ag(I) were optimal for this reaction. To delve deeper into this phenomenon, we conducted several controlled experiments, as outlined below:

*Entry	Oxidant	NMR Yield (%)
1	Ag ₂ CO ₃	61
2	4.0 equiv. Ag ₂ CO ₃	58
3	Ag ₂ CO ₃ under argon atmosphere	trace
4	Ag ₂ CO ₃ under nitrogen atmosphere	28%
5	4.0 equiv. Ag ₂ CO ₃ under nitrogen atmosphere	44%

(*It is only a comparison table using different environment. For a detailed and full oxidant optimization studies, please see, Supporting Information, **Table S7** and **Table S8**, page no. 8-10).

From the oxidant optimizations, it is evident that Ag(I) salt alone is not effective as an oxidant in this particular transformation. Notably, experiments conducted under inert atmosphere revealed that even a super stoichiometric amount of silver carbonate failed to provide similar yield (Entry 3, 4, 5) compared to the reaction using 2 equiv. of Ag₂CO₃ in air (Entry 2). Hence, it is reasonable to conclude that the synergistic action of Ag(I) and aerial oxygen serves as an oxidant system in this reaction. Such observation is also supported by the literature (Izawa *et al. Science*, **333**, 209-213 (2011)), where dehydrogenation / cyclic ring aromatization was facilitated using Pd-catalyst in combination with O₂ oxidant. Accordingly, we have specified 'air' in each of the reaction schemes as an oxidant system along with Ag₂CO₃ in the MS and SI. Additionally, detailed discussions regarding the role of atmospheric oxygen have been provided in the supplementary information section 2 (page no. 10).

Furthermore, in response to the reviewer's helpful suggestion, we conducted an investigation into the involvement of Pd active species from Pd-hydride. To do so, we carried out a controlled experiment involving an excess of an external alkene under an inert gas atmosphere.

Initially, we observed that conducting the reaction under an inert atmosphere resulted in a decrease in yield, achieving only 28% of the product (please refer to the comparison table mentioned above). However, upon introducing norbornene as a Pd-hydride acceptor, the yield of the current reaction further decreased, dropping drastically to 8%. making it unfeasible for the catalytic cycle to continue. Hence, we conclude that excess alkenes cannot serve as Pd-hydride acceptor. Nevertheless, the olefination at the β -site did contribute to accelerating the outcome of the catalytic cycle, as also detailed in the main manuscript: “*Noteworthy, the low yield observed for the formation of arene products without the olefin coupling partner suggests that the olefination accelerates the decarboxylative aromatization.*” Further detailed mechanistic investigation supported with DFT studies are currently underway in our laboratory.

Reviewer’s comment 2: Scheme 5: The authors indicated that the oxidation proceeded site-selectively by showing decarboxylation site and acid addition site separately. I am afraid that is clearly mis-leading readers. If the reaction proceeds via symmetric π -allyl Pd, then, both sites should be oxidized. Thus, it is impossible to differentiate decarboxylation site and acid addition site.

Our response: We thank the reviewer for helpful suggestion. We do agree with the reviewer that, in Scheme 5, there is equal probability of acyloxylation from both the sites of the π -allylic complex. However, acyloxylation from either way eventually would generate the same products (due to symmetric cyclopentene- π -allylic intermediate). The plausible mechanism for Scheme 5, which was previously presented in the supplementary information, has now been incorporated into the main manuscript as Scheme 8 (b) with slight modifications and highlighted in yellow. Pd-catalyzed allylic-acyloxylation from both the sites of the π -allylic intermediate has been demonstrated. A detailed description of this mechanism has been included in the supporting information (Section 8. e, page no. 78-79).

Reviewer's comment 3: In SI, information on starting materials may be missing. Did authors use only commercially available carboxylic acids as described in page 3?

Our response: We thank the reviewer for the careful review. Most of the starting materials are commercially available and bought from various sources (Johnson Matthey, BLDpharm, Chempure, Zeta Scientific, Aldrich and TCI-India). Starting material acids, which are synthesized, have been listed in the revised SI. The synthetic routes along with the characterization data have been included in the supplementary information (page no. 15-22).

Reviewer's comment 4: *d.r.* of products 10 and 11 are not reported.

Our response: We are thankful to the reviewer for thorough evaluation. Product **10** and **11** do not have any *d.r.*, as the starting material acids are themselves chiral (which are commercially available). The HPLC traces of starting materials are provided below.

For, *trans, trans*-4'-ethylbicyclohexyl-4-carboxylic acid

Enantiomeric ratio was determined by HPLC with a Daicel Chiralpak AD-H, n-hexane/2-propanol = 99.5/0.5, $v = 1.0 \text{ mL}\cdot\text{min}^{-1}$, $\lambda = 270 \text{ nm}$, $t = 8.167 \text{ min}$.

For, *trans, trans*-4'-pentylbicyclohexyl-4-carboxylic acid

Enantiomeric ratio was determined by HPLC with a Daicel Chiralpak AD-H, n-hexane/2-propanol = 99.5/0.5, $v = 1.0 \text{ mL}\cdot\text{min}^{-1}$, $\lambda = 270 \text{ nm}$, $t = 8.413 \text{ min}$.

In the revised manuscript and Supplementary Information (SI), we have updated the structures of these products and provided specific details regarding their stereochemistry.

Previously in MS and SI:

Now in revised MS and SI:

Additional changes:

1. In the introductory scheme, product of Shapiro reaction was drawn incorrectly. We have rectified the unwilling mistake in this revised version.
2. In the previous manuscript, we observed that the NMR spectra of one product (previously listed as Entry **78**) contained a notable amount of impurities. Our initial intention was to re-purify the product; however, despite our efforts, we were unable to achieve success in doing so. Consequently, we have made the decision to remove the entry for that product (Entry **78**) and have replaced it with a newly synthesized product, now designated as Entry **80**, which aligns with the corresponding table in the revised manuscript.
3. With the addition of two new entries (**23**, **24**), the numbering of the rest of the entries changed. We have made the changes accordingly in the MS and SI with highlighting those in **yellow background**. Also, we have added multiple citations in this revised MS and SI and so, highlighted the changes in the "References" section.
4. As mentioned by the editor, we've incorporated a "**Data Availability**" section and relocated the "**Methods**" section after the conclusion, as instructed.
5. The previous version of this manuscript and supplementary information contained a few typos and grammatical errors, which have been corrected in the revised versions.

We have made our best efforts to address the queries. I sincerely hope that you will find this revised manuscript suitable for publication in *Nature Communications*.

Sincerely,

Debabrata Maiti

REVIEWERS' COMMENTS

Reviewer #1 (Remarks to the Author):

This revised manuscript has addressed my previous concerns, thus I recommend accepting the manuscript, except some literature format need to be corrected.

Reviewer #2 (Remarks to the Author):

The authors have provided satisfactory responses to the comments. I recommend proceeding with the publication of the manuscript.

Reviewer #3 (Remarks to the Author):

The authors revised the manuscript in responses to all the reviewers. My previous concern, especially lack of enough oxidant for the proposed catalytic cycle, was clarified now with additional information provided (the reaction was performed under air). It is reasonable that O₂ assisted the regeneration of active Pd species.

Other scientifically unclear points were also improved and so the manuscript is now suitable for publication.

Response Letter

Reviewer 1

Reviewer #1 (Remarks to the Author): This revised manuscript has addressed my previous concerns, thus I recommend accepting the manuscript, except some literature format need to be corrected.

Our response: We are thankful to the reviewer for favorable recommendation in accepting the manuscript. The literature format has been adjusted as per the editor's guidelines in the revised manuscript, with changes highlighted using a yellow background.

Reviewer 2

Reviewer #2 (Remarks to the Author): The authors have provided satisfactory responses to the comments. I recommend proceeding with the publication of the manuscript.

Our response: We are extremely grateful to the reviewer for the kind appreciation and supporting the publication of this work in *Nature Communications*.

Reviewer 3

Reviewer #3 (Remarks to the Author): The authors revised the manuscript in responses to all the reviewers. My previous concern, especially lack of enough oxidant for the proposed catalytic cycle, was clarified now with additional information provided (the reaction was performed under air). It is reasonable that O₂ assisted the regeneration of active Pd species.

Other scientifically unclear points were also improved and so the manuscript is now suitable for publication.

Our response: We are thankful to the reviewer that he/she finds the manuscript suitable for publication in *Nature Communications* after revision.

I sincerely hope that you will find this revised manuscript suitable for broad readership of *Nature Communications*.

Sincerely,

Debabrata Maiti